# Histone Post-Translational Modifications and CircRNAs in Mouse and Human Spermatozoa: Potential Epigenetic Marks to Assess Human Sperm Quality

**DOI:** 10.3390/jcm9030640

**Published:** 2020-02-27

**Authors:** Teresa Chioccarelli, Riccardo Pierantoni, Francesco Manfrevola, Veronica Porreca, Silvia Fasano, Rosanna Chianese, Gilda Cobellis

**Affiliations:** Department of Experimental Medicine, University of Campania “L. Vanvitelli”, Via Costantinopoli 16, 80138 Napoli, Italy; teresa.chioccarelli@unicampania.it (T.C.); riccardo.pierantoni@unicampania.it (R.P.); francesco.manfrevola@unicampania.it (F.M.); veronica.porreca@unicampania.it (V.P.); silvia.fasano@unicampania.it (S.F.); gilda.cobellis@unicampania.it (G.C.)

**Keywords:** histone post-translational modifications (histone PTMs), circular RNAs (circRNAs), human male infertility, sperm, spermatogenesis, fertilization, embryo development

## Abstract

Spermatozoa (SPZ) are motile cells, characterized by a cargo of epigenetic information including histone post-translational modifications (histone PTMs) and non-coding RNAs. Specific histone PTMs are present in developing germ cells, with a key role in spermatogenic events such as self-renewal and commitment of spermatogonia (SPG), meiotic recombination, nuclear condensation in spermatids (SPT). Nuclear condensation is related to chromatin remodeling events and requires a massive histone-to-protamine exchange. After this event a small percentage of chromatin is condensed by histones and SPZ contain nucleoprotamines and a small fraction of nucleohistone chromatin carrying a landascape of histone PTMs. Circular RNAs (circRNAs), a new class of non-coding RNAs, characterized by a nonlinear back-spliced junction, able to play as microRNA (miRNA) sponges, protein scaffolds and translation templates, have been recently characterized in both human and mouse SPZ. Since their abundance in eukaryote tissues, it is challenging to deepen their biological function, especially in the field of reproduction. Here we review the critical role of histone PTMs in male germ cells and the profile of circRNAs in mouse and human SPZ. Furthermore, we discuss their suggested role as novel epigenetic biomarkers to assess sperm quality and improve artificial insemination procedure.

## 1. Glossary[Up3] [AG4] 

### 1.1. Histone Acetylation

Histone acetylation mainly occurs on the N-terminal tails of the histone core (H2A, H2B, H3 and H4). The enzymatic process is catalyzed by histone acetyl transferase (HAT), which requires acetyl-CoA as coenzyme for the transfer of acetyl group (-COCH3) to the lysine acceptor, while-COCH3 removal is catalyzed by histone deacetylase (HDAC). HATs and HDACs are members of a large enzyme family.

Typical lysine (K) acetylation sites are K5, K8, K12, K16 and K20 of histone H4; K9, K14, K18 and K23 of histone H3; K5, K12, K15 and K20 of histone H2B; K5 and K9 of histone H2A.

Histone acetylation is primarily (not always) linked to transcriptionally active chromatin, whereas histone diacetylation leads to inhibition of transcriptional activity.

i.e.,: Histone H4 acetylated in lysine 16 is designated as H4K16ac.

### 1.2. Histone Methylation

Histone methylation occurs mainly on H3 and H4 tails and involves the transfer of methyl group (CH3) from the enzymatic donor SAM to lysine (K) or arginine (R) acceptor. Lysine residues can be mono-di- and tri-methylated. Arginine residues can be mono- and di-methylated. The enzymatic process is catalyzed by histone methyltransferase (HMT) whereas CH3 removal is catalyzed by histone demethylases (HDM). HMTs and HDMs are members of a large enzyme family.

Histone lysine methylation is associated with both transcriptionally active chromatin (H3K4, H3K36, H3K79) and transcriptionally repressive chromatin (H3K9, H3K27, H4K20). The methylation of H3K9/K27 is involved in the formation of heterochromatin. 

Arginine methylation usually occurs on H3R2, H3R8, H3R17, H3R26, H4R3, H2AR3. The association between H3R17 and H4R3 is typically linked to active chromatin whereas the association of H3R8 and H4R3 is linked to repressive chromatin.

i.e.,: Histone H3 (mono-di- or tri-) methylated in lysine 4 is designated as H3K4me(-1-2 or -3) and Histone H4 methylated in arginine 3 is designated as H4R3me.

### 1.3. Histone Phosphorylation

Histone phosphorylation describes the addition of phosphate group (PO4) to histone tails at serine, threonine and tyrosine residues. The enzymatic process is catalyzed by histone kinase while-PO4 removal is catalyzed by histone phosphatases. The negative charges associated to PO4 influence the chromatin structure. Histone phosphorylation controls several processes, including mitosis, meiosis and DNA damage response.

*Example*: Histone H4 phosphorylated in serine 1 is designated as H4S1ph.

### 1.4. Histone Ubiquitination 

Ubiquitin is a small protein. Its covalent addition to target proteins requires E1–E3 enzymes. Ubiquitination is specifically catalyzed by sequential activity of three enzymes: ubiquitin is activated by ubiquitin-activating enzyme (E1), subsequently transferred to ubiquitin-conjugating enzyme (E2) and recognized by the ubiquitin-protein ligase (E3), which catalyzes the transfer of the ubiquitin to the substrate proteins at lysine residues.

The H2A and H2B histones are preferential targets of mono-ubiquitination. Histone H2B ubiquitination seem to be linked to transcriptional activation as it facilitates H3K4 methylation, whereas ubiquitination of histone H2A at lysine 119 is related to transcriptional repression. 

i.e.,: Histone H2A ubiquitinated in lysine 119 is designated as H2AK119ub.

### 1.5. Histone SUMOylation

Small ubiquitin-related modifier polypeptide SUMO is a small protein. Different SUMO proteins have been described in mammals: SUMO-1, SUMO-2, SUMO-3 and SUMO-4. Like ubiquitination, its covalent addition to target proteins, namely SUMOylation, is specifically catalyzed by sequential activity of E1-E3 enzymes: SUMO is activated by the E1-activating enzyme (a heterodimer of SAE1/SAE2), then is passed to the active site of the E2-conjugating enzyme (UBC9) which catalyzes, in association to E3 ligase, the conjugation of SUMO to substrates by the formation of an isopeptide bond. Sumoylation has been observed at lysine residues of histone H2A (K126), H2B (K6, K7, K16, and K17), and H4 (K5, K8, K12, K16 and K20).

### 1.6. Histone Crotonylation

Crotonylation is a new post-translational modification (PTM) of histone tails in which the crotonyl group (C4H5O) is transferred from crotonyl-CoA to the lysine target residue (K-cr). Histone crotonylation and decrotonylation are specifically regulated by histone crotonylase and decrotonylase. Interestingly, histone acetylase and deacetylases may efficiently act as histone crotonylase and decrotonylase, respectively. Crotonylation promotes transcriptional activity as it opens the chromatin structure. Indeed, it is present in the promoter and enhancer regions.

### 1.7. Histone Poly-ADP-Ribosylation

ADP-ribosylation is a reversible PTM in which the ADP-ribose is transferred from the nicotinamide adenine dinucleotide (NAD^+^) substrate to amino acid acceptor. However, mono-ADP-ribosylation especially concerns proteins other than histones, whereas poly-ADP-ribosylation mainly occurs against nuclear proteins. Poly-ADP-ribosylation is catalyzed by poly ADP-ribose polymerases (PAR) (mainly PARP1 and PARP2) and reversed by PAR glycohydrolase (PARG). 

## 2. Introduction

The large storage of material in the oocyte and the extensive chromatin condensation in spermatozoa (SPZ) are events specifically characterizing female and male gametes, respectively. Up to 10 years ago the widely accepted idea was that such events reflect the different aims of these two cells: to orchestrate the entire post-zygotic developmental process (oocyte) and to provide an undamaged genome to fertilization (SPZ). However, SPZ are more than mere vectors for conveying the paternal haploid genome to the oocyte. Several heritable diseases or some life experiences able to induce phenotypic alterations in the offspring are not sufficiently explained just through DNA sequence-based inheritance. A new wave of research has been stimulated to explore this “missing heritability problem”, pointing to the epigenetic information as the joining link between the environment and the genetics. Indeed, epigenetics (i.e., above genetics) is referred to as the “study of heritable changes in gene expression that occur without changes in DNA sequence” [1]. In recent years, a number of studies have pointed towards sperm epigenetic information as a potential contributory factor to embryonic development as well as offspring’s health [2]. It is now clear that SPZ transmit a highly dynamic and environmentally susceptible epigenetic code to oocytes, essential for both fertilization and normal embryonic development [3,4]. In this context, DNA methylation has been the most studied epigenetic modification. However, histone PTMs and non-coding RNA (ncRNA) activity are gaining considerable attention.

Histone PTMs have been characterized in germ cells, including mouse and human SPZ, and their key function in epigenetic regulation of embryo development is emerging [5], emphasizing their role in sperm quality. Nevertheless, a large cargo of ncRNAs enriches SPZ. Several lines of evidence point to ncRNAs as involved molecules in sperm quality control. The recent discovery of circular RNAs (circRNAs)—a subclass of ncRNAs—in mouse and human SPZ and their suggested role in gene expression regulation make these molecules novel epigenetic biomarkers to assess sperm quality.

In this review we will focus on the critical role of histone PTMs in mouse germ cells during spermatogenesis and describe histone PTMs and circRNA profile in mouse and human SPZ. Furthermore, we discuss their suggested role as novel epigenetic biomarkers to assess sperm quality in order to improve artificial insemination procedure.

### 2.1. Histone Code and Histone PTMs

Chromatin is a highly condensed histone-DNA complex. The nucleosome is the fundamental repeating subunit of chromatin: eight histones (two each of H2A, H2B, H3 and H4) organize the histone core around which DNA rolls up making a double turn (about 146 bp). Histone H1 interacts with nucleosomal DNA and—[AG5] by bridging adjacent nucleosomes—it condenses DNA-linker and stabilizes higher-order chromatin structure. Histones are small, highly basic proteins with unstructured N-terminal and C-terminal ends, namely histone tails, that become positively charged due to being rich in basic amino acids, such as arginine and lysine. The octameric histone core superficially exposes the histone tails. This facilitates the histone-DNA interaction, DNA being negatively charged. Regardless of the structural character, histones are critical regulators of the dynamic state of chromatin with functional implications. They directly participate in many different DNA-based mechanisms, including transcription [6]. Such a role is primarily related to the activity of histone-modifying enzymes that covalently add or remove PTMs at multiple sites-lysine (K), arginine (R), serine (S), threonine (T)-localized at the extended N-terminal tails. Numerous PTMs, including acetylation (ac), methylation (me), ubiquitylation (ub), phosphorylation (ph), SUMOylation (sumo), ADP-ribosylation, crotonylation (cr), have been characterized (see glossary). Functional significance of histone PTMs, i.e., histone code, depends on both specific aminoacidic positioning and combination of histone PTMs. Overall, histone PTMs significantly reshape the local chromatin conformation since they affect stability of the histone core or histone-DNA interactions. For example, lysine acetylation of histone tails reduces the number of positive charges and weakens the histone-DNA interaction facilitating chromatin accessibility. Consequently, the increased acetylation is mainly associated with activation of gene expression. This occurs habitually but not always. Histone methylation may have positive or negative effects on gene expression depending on the specific amino acid residue and histone type modified. Increasing findings propose that histone PTMs may act sequentially or in combinatorial way, involving one or more histone tails, to form the histone code that is deciphered by docking proteins that trigger distinct downstream biological effects. Indeed, modified histones recruit chromatin remodeling complexes [7] and/or transcription factors [8]. Some co-activators and co-repressors associated with transcription factors are histone-modifying enzymes [9]. The emerging idea is that a number of multiprotein complexes, differently combined, and referable to unique machinery may operate to remodel chromatin and affect gene expression and chromatin packaging [10].

Many histone PTMs and related histone-modifying enzymes have been described in testis, primarily in mouse germ cells (Table 1, refs. [11,12,13,14,15,16,17,18,19,20,21,22,23,24,25,26,27,28,29,30,31,32,33,34,35]). Histone PTMs have been further characterized in germ cells, including mouse [36] and human [37] SPZ, where at least 30 different modifications have been reported [36]. These have key roles in chromatin and DNA-based mechanisms occurring in germ cells during spermatogenesis. Their function in epigenetic regulation of fertilization and embryo development is emerging [38]. Until recently, the sperm histone-mediated epigenetic information has been poorly considered, because of chromatin remodeling events related to SPT (i.e., histone-to-protamine exchange) [39], and erasing of some histone PTMs observed early after fertilization, before zygote genome activation [40]. However, SPZ retain a small fraction of histones that become modified by PTMs and non-randomly localized. 

### 2.2. Sperm RNA Code

Beyond DNA methylation, histone-to-protamine replacement and histone PTMs, a large cargo of RNAs enriches SPZ. 

SPZ-borne RNAs constitute a heterogeneous family of both coding and ncRNAs [41]. According to their size, ncRNAs may be grouped in long or small ncRNAs [42,43]. Among the small ncRNAs, tRNAs, small nucleolar and nuclear RNAs have been identified as the small *housekeeping* ncRNAs, instead PIWI-interacting RNAs (piRNAs) and microRNAs (miRNAs) are known as small *regulatory* ncRNAs. In addition to miRNAs and piRNAs [44,45], tRNA-derived small RNAs (tsRNAs) populate mature SPZ [46,47]. These are 5′ fragments of tRNAs with a size from 29 to 34 nt, whose biogenesis still remains largely unknown. According to the region on tRNAs from which they are derived, tRNAs can be classified into five groups: 5′-tRNA halves, 3′-tRNA halves, 5′-tRFs (tRNA-derived RNA fragment), i-tRFs (internal tRFs) and 3′-tRFs [48]. In human SPZ, 75% of all tRNAs are represented by 5′-tRNA halves [49]. The discovery of tsRNAs in SPZ has suggested the potential existence of an underestimated ‘housekeeping RNA-derived’ small RNA family. Actually, both human and mouse SPZ have been shown to also contain an appreciable amount of ribosomal RNA (rRNA)-derived small RNAs (rsRNAs)-produced by the cleavage of rRNAs, a necessary event in order to ensure the translational shutdown in mature SPZ-whose abundance increases along the epididymis where SPZ transit and mature [50]. As for tRNAs, molecular mechanisms for rsRNA biogenesis remain to be explored. However, according to the subtypes of rRNA precursors (5S, 5.8S, 18S, 28S, 45S) from which they are derived, rsRNAs can be divided into five types, with 60% of all rsRNAs represented by 28S rsRNA in human sperm [49]. To date, a tRNA methyltransferase, DNMT2, has been linked to both tsRNA and rsRNA biogenesis in sperm, since *DNMT2* knockout mice have altered composition of these two classes of ncRNAs [51]. 

In addition to the sequence diversity of sperm RNAs, ncRNA landscape harbors a plethora of RNA modifications, especially in tsRNA and rsRNAs, probably because they derive from RNAs rich in chemical modifications, such as tRNAs and rRNAs [52]. In general, RNA modifications increase RNA stability and prolong their function in cells. Among modifications, 5-mehylcytosine (m^5^C) and *N*1-methyladenosine (m^1^A) are very common; they contribute to the potential binding between RNAs and specific proteins or they generate unexpected secondary RNA structures during the folding [53,54]. 

Therefore, the wide family of ncRNAs in SPZ and the high complexity due to the huge of associated RNA modifications are expanding, constituting a real “*sperm RNA code*” with many aspects still to be deciphered.

Interestingly, in sperm, RNAs can be localized in main compartments: (i) extra-nuclear compartment that includes the plasma membrane, the acrosome and associated membranes; (ii) the intra-nuclear compartment which includes the nucleus, the nuclear envelope and the perinuclear theca; (iii) mitochondria. Most mRNAs and rsRNAs are attached to the sperm outer membrane [55] and a possible exocytosis of these molecules in extracellular vesicles has been hypothesized [56]. On the other hand, tsRNAs and miRNAs appear imbedded in the nuclear envelop; it cannot be ruled out that a population of RNAs may directly contact DNA forming highly structured RNA-DNA complexes [57]. Sperm tail is also rich in tsRNAs, miRNAs and piRNAs [47]. Sperm mitochondria may constitute a site of active transcription in transcriptionally silent cells [58]; however, such a hypothesis remains to be experimentally verified.

NcRNA production firstly occurs during spermatogenesis when several events need to be regulated. However, ncRNAs in SPZ are unlikely to be just the remnants of spermatogenesis; they represent an exogenous cargo delivered to SPZ during their maturation (Table 2, refs. [59,60,61,62,63,64,65,66,67,68,69,70,71,72,73,74,75,76,77,78,79,80], outlines the main functions of sperm RNA types in spermatogenesis and SPZ). 

Since SPZ are transcriptionally and translationally silent cells before departing the testes [81], it is plausible that the substantial remodeling of their ncRNA cargo may be driven extrinsically along the epididymis, a long convoluted tubule connecting rete testis to the vas deferens, composed of three main anatomical regions (*caput*, *corpus* and *cauda*), with a considerable segment to segment variation. Leading candidates to promote this crosstalk are epididymal epithelial cells that communicate with SPZ through epididymosomes, vesicles—heterogeneous in their lipid/protein/RNA composition and density—released through an apocrine secretory mechanism [82]. Epididymosome-sperm interactions are likely selective and involve specialized membrane micro-domains known as lipid rafts [83]; furthermore, a transient fusion pore could be created on sperm membrane to allow the delivery of the cargo and several trafficking proteins could mediate this intercellular communication [84]. Particularly intriguing is that molecular mechanisms on the basis of this communication are largely unknown. Interestingly, epididymosome collection along the length of the epididymis has revealed a segment to segment variation in ncRNA content [85], with a decrease in miRNA and a substantial enrichment in piRNA/tsRNA content in SPZ collected from cauda epididymis [44,85].

Nowadays, ncRNAs are molecules involved in sperm quality control. Interestingly, a recent study suggests a differential expression of tsRNAs, rsRNAs and miRNAs in sperm samples collected from male partners of couples undergoing in vitro fertilization (IVF) in relation to embryo quality, even though these sperm samples were all considered normal by traditional semen-parameter assessment [49]. This evidence highlights the limitation of the morphological analysis and the necessity to evaluate the epigenetic quality of SPZ. Aberrant miRNA expression profiles have been associated to patients with oligozoospermia, revealing a fingerprint of impaired spermatogenesis [86]. The inhibition of sperm-delivered miRNAs in mice results in developmental delay in the zygote [65]. Additionally, sperm derived from knock-out mice for *DICER*/*DROSHA*, the enzymes involved in miRNA maturation, when used to fertilize oocytes, induce a deregulation in the expression of the embryo preimplantation genes. Such a phenotype can be recovered by injecting a pool of sperm ncRNAs from wild-type mice [87]. Furthermore, the ncRNA cargo gained during epididymal transit is essential for sperm quality and embryo development, as a consequence. In fact, immature caput sperm generates embryos with signatures of aberrant preimplantation. The co-injection of embryo with ncRNA cargo extracted from cauda epididymosomes improves such a phenotype [4].

A subclass of ncRNAs-circRNAs-discovered in mammalian cells in 1979 has a very debated history, due to its low abundance and uncommon features, thus to be considered a byproduct of abnormal splicing [88]. Ten years later, circRNA existence has been shown in human cancerous cells, but the biological role of these molecules still remained unclear [89]. With the development of high-throughput sequencing technologies and bioinformatic approaches, circRNA research has had a break-through. Recent literature based on RNA sequencing data estimates that circRNAs may represent the largest RNA family in human transcriptome, especially in brain and testis [90], thus to open up new routes in ncRNA field.

The relationship between epigenetic factors and male infertility is still at its starting point. The existence of an RNA code, with a high degree of versatility and multiple layers of information (sequence, modifications, folding), introduces new challenges in reproductive biology field. Even more interesting is the ability of sperm RNA code to interact with environmental, genetic and other epigenetic factors in order to influence cell quality and alter the epigenetic signature of the offspring [91]. Figure 1 shows a schematic view of the epigenetic mechanisms with a focus on the main classes of ncRNAs found in SPZ.

However, in such a scenario, the research on the role of circRNAs, in comparison to the other described ncRNAs, is still at its childhood.

## 3. Histone PTMs Related to Spermatogenetic Progression of Germ Cell in Mice

Spermatogenesis is a sex specific differentiation program highly conserved in vertebrates [92]. This is a long-life continuous process as spermatogonial stem cells maintain homeostasis of spermatogenesis. Phases characterizing germ cell progression are: (i) self-renewal of spermatogonial stem cells, commitment and proliferation of spermatogonia (SPG); (ii) meiotic entry, progression and production of haploid round SPT; (iii) morphogenesis of round SPT into SPZ. The histomorphological reshaping of round SPT, also known as spermiogenesis, includes one of the hallmarks of spermiogenesis, i.e., the extreme chromatin condensation. Such event greatly reduces sperm head size and increases hydrodynamic shape of SPZ [93].

Histone PTMs regulate germ cell progression as primarily described in mouse testis. Figure 2 is a schematization of it. 

### 3.1. Self-Renewal, Commitment and Proliferation of SPG

In mice, self-renewal of spermatogonial stem cells produces undifferentiated SPG (i.e., single, paired, aligned SPG) which differentiate (commitment phase in type A1 SPG) and proliferate (A1–A4 SPG, intermediated and type B SPG) before starting meiosis as spermatocytes (SPC). The positive or negative expression of *c-KIT* gene categorizes SPG. Indeed, the single, paired, aligned SPG show stem cell activity and become *c-KIT* negative cells. Starting from A1, differentiating SPG become *c-KIT* positive cells, with c-kit being necessary for migration, proliferation, and differentiation of type-A SPG [94,95]. The self-renewal of spermatogonial stem cells maintains homeostasis of spermatogenesis through the expression of a specific set of genes [96]. Many of these are transcriptionally active and others repressed with the purpose to create a definite transcriptional profile of gene expression. Histone PTMs play an important role in this context. In particular, histone acetylation and methylation, generally involved in chromatin activation and repression, respectively, are the most common histone PTMs in SPG cells [97,98,99]. Interestingly, this opposed function, i.e., transcriptional activation and repression, leads to changes in gene expression. Furthermore, the histone PTMs-based chromatin reorganization might function as a switching point that governs self-renewal and differentiation/commitment.

While histone methylation plays a critical role in self-renewal maintenance of spermatogonial stem cells, to date there is little information about the role of acetylation on the control of this activity. Studies in vitro on undifferentiated SPG show that H4K8ac and H4K16ac are accumulated around transcription start sites (TSSs) of constitutively active genes. Interestingly, in post-meiotic germ cells, these modifications are less localized on inactive genes that become highly expressed in undifferentiated SPG [100].

Maintenance of spermatogonial stem cells, requires the promyelocytic leukaemia zinc finger transcriptional repressor, PLZF. The PLZF-expressing undifferentiated SPG preferentially have H3K27, H3K9 and H4K20 in their di- and trimethylated states [101], imparting an epigenetic silencing more-lasting than the mono-status [102]. This expression pattern, in turn, could reflect a higher degree of chromatin silencing to ensure an undifferentiated cellular state [101].

Typically, heterochromatin is not present in undifferentiated SPG [103], while it appears as SPG differentiate [104]. Heterochromatin harbors transcriptionally repressive histone tail modifications, such as H3K9me2 [105,106]. Noteworthy, H3K9me2 is linked to facultative heterochromatin, whereas H3K9me3 is linked to constitutive heterochromatin. Indeed, it provides a binding site for heterochromatin protein-1 (HP1) essential for heterochromatin formation [107]. Consistently, undifferentiated SPG lack the repressive H3K9me2 transcriptional tag, while they show H3K9me3 at the nuclear periphery. When undifferentiated SPG lose stem cell activity and start differentiation/commitment (transition from aligned SPG-to-A1SPG), they acquire H3K9me2 and redistribute H3K9me3 at the nuclear level. Indeed, H3K9me3 appears to DAPI-dense foci [108]. Accordingly, methylation of H3K9, by G9a, blocks gene expression of the stem cell factors OCT4 and NANOG, therefore demethylation of H3K9, by JMJD1C, is a significant step in the maintenance of self-renewal of spermatogonial stem cells [12,20] (Figure 3A,B). However, if on one hand methylation of H3K9 must be removed to allow gene expression of stem cell factors as OCT4 and NANOG, on the other methylation of H3K9 is fundamental to block the expression of genes that could compromise their survival. Indeed, it has been shown that H3K9me3 regulates survival of spermatogonial stem cells by repressing gene expression of apoptotic genes [17].

Accumulating evidence considers H3K4me2 in the maintenance of transcriptional active states. Interestingly, conditional deletion of *KDM1A*, a H3K4me2 demethylase, induces a severe effect on differentiation of SPG and a complete loss of germ cells [22,109]. Remarkably, *KDM1A* loss is associated with a profound alteration of gene expression including those that are essential to undifferentiated SPG maintenance (i.e., *PLZF*, *NANOS2* and *BCL6b*) [109]. The KDM1A has been shown to directly bind *OCT4* gene in a mouse germ cell line (GC-1 cells) and remove H3K4me2 at the promoter and proximal enhancers. This allows transcriptional repression of *OCT4*, favoring SPG commitment and differentiation [22] (Figure 3A,B).

During mouse spermatogenesis, SPG are enriched of acetylated histones, particularly H3 and H4 [97,98]. H2A and H2B acetylation is also detected in mouse type-A and type-B SPG and could be essential for histone deposition and displacement during DNA replication of proliferating SPG [110]. Several histone H3 acetylations are detected in type-A SPG and type-B SPG, including H3K9ac, H3K18ac and H3K23ac [98].

Histone H3K4, H3K9, H3K27 and H4K20 methylation turns out to be of strong epigenetic impact for SPG differentiation. Particularly, H3 modification linked to transcriptional activation results in more type-B SPG as compared to type-A SPG, suggesting a new gene expression profile occurs in the differentiation phase [99]. Accordingly, H3K4 mono-, di-, and tri-methylation are detected in mouse testis at 6 and 10 *days post-partum* (*dpp*), corresponding to the appearance of type-A to type-B SPG; H3K4me3 gradually increases in type-B SPG to induce the expression of genes that regulate cell differentiation and meiotic entry [99]. Conversely, H3K27me3, linked with repression of transcriptional activity, results more present then H3K4me3 in type-A SPG to favor the silencing of gene probably involved into self-renewal maintenance. Bivalent signature H3K4me3/H3K27me3 is present in promoter regions of pluripotency factors such as *NANOG*, *SOX2*, *LEFTY* and *PRMD14*. The lacked expression of these factors in spermatogonial stem cells regulates self-renewal and proliferation [111]. Noteworthy is that the loss of H3K27me3 from bivalent signature in promoter of genes such as *ALDH2*, *STRA8* and *SPO11* induce chromatin activation that favors SPG differentiation and meiotic entry [111].

### 3.2. Meiosis

Self-renewal, commitment and proliferation produce a number of type-B SPG that differentiate and enter in meiosis as preleptotene (PL) SPC (_PL_SPC). In mice, this occurs around 10–12 dpp. Typical events of meiosis are condensation, synapsis, homologous recombination and segregation of chromosomes as well as sex chromosome inactivation. Peculiarly, early SPC are transcriptionally active cells. RNA synthesis is highest during the mid-pachytene phase and the end of the prophase I [92].

Histone PTMs regulate many meiotic events such as transcriptional regulation, recombination and DNA double strand breaks (DSBs) repair as well as meiotic sex chromosome inactivation (MSCI).

During prophase I, a significant transcriptional activation occurs in leptotene (L) and zygotene (Z), while it decreases in pachytene (P). Several histone modifications support this transcriptional activation. In particular, acetylations of histone H3 (H3K9ac, H3K18ac, H3K23ac) and H4 (H4K5ac, H4K8ac, H4K12ac) are strongly present in L–Z and less in P when the rapid transcriptional inactivation is instead supported by histone H3 and H4 methylation (H3K27me3, H4K20me3) [98,112]. 

In mice and humans, meiotic recombination initiates through the site-specific activation of recombination. Interestingly, histone H4 acetylation (H4K5ac, H4K8ac, H4K12ac, H4K16ac) is enriched in PL-SPC to promote chromatin opening to facilitate formation of the recombination hotspots (Figure 4A), confirming the data observed in *Saccharomyces cerevisiae* [113,114,115].

Activating histone methylation marks (H3K4me2, H3K36me3, H3K79me1) are generally present on hotspots, while repressive histone methylation marks (H3K27me3, H3K9me3, H4K20me3) are absent, facilitating chromatin opening. Repressive histone methylation marks are present in chromatin regions correlated with reduced recombination or with heterochromatin formation, and are not involved in hotspots formation [113,116]. Recombination hotspots are marked by H3K4me3 induced by a DNA-binding zinc-finger protein with histone tri-methyl transferase activity named PRMD9 or Meisetz, expressed in L and Z-SPC [15,117,118,119]. Recently, it has been demonstrated that PRMD9 can tri-methylate H3K36 in mouse SPC, coupling it with H3K4me3 on the same nucleosome thus to create a bivalent signature, exclusively in the region of recombination [120]. H3K4me3 on hotspots core facilitates chromatin opening by the endonuclease SPO11 action and DSBs formation (Figure 4B,C). 

H4S1ph and H2AS1ph localize on DSBs and are correlated to meiotic recombination [121,122]. 

H3K9ac and H3K14ac together act as bivalent signature that promotes access to the repair machinery on DSBs whereas H4K91ac is involved in chromatin assembly of repair machinery [123]. 

Homologous recombination (HR) is the main DSBs repair pathway that occurs from L-to P-SPC. In this context, the phosphorylation of S139 of histone H2AX (i.e., γH2AX) is fundamental. In L, this phosphorylation is ATM-dependent, whereas in Z it is ATR-dependent (Figure 4D). The two waves of H2AX phosphorylation are involved in the recruiting of HR repair factors such as RAD51 for the correct repair of DSBs and for an efficient meiotic recombination [124,125]. Interestingly, ubiquitination of histone H2B, catalyzed by RNF20, also regulates meiotic recombination. In fact, H2BK120ub is involved in chromatin relaxation necessary for the recruitment of DSBs repair factors, such as RAD51, MDC1 and BRCA1, promoting a correct homologous recombination [28,126] (Figure 4E). Both acetylation and methylation of histone H4 are particularly important in the DSBs repair mechanism. Germ cell-specific deletion of histone deacetylase MOF causes global loss of H4K16ac [25,97]. This event limits the phosphorylation wave of H2AX in L and Z and negatively affects DSBs repair and crossover numbers [25]. Similarly, the genetic loss of the arginine methyltransferase Prmt5 induces a meiotic block related to H4R3me2 reduction and abnormal presence of multiple γH2AX foci, a sign of incorrect DSBs repair during meiosis [127]. Interestingly, γH2AX is linked to MDC1 to favor HR repair (Figure 4E). 

During meiosis, in L-SPC, the X and the Y chromosomes are randomly localized in the nucleus. When recombination starts in P-SPC, the sex chromosomes migrate toward the periphery of the nucleus, called “XY body”, to separate them from autosomes. In this phase, whereas autosomes totally synapsed and start HR, sex chromosomes pair only in their “pseudo autosomal regions”. The non-complementary regions of sex chromosomes remain asynapsed leading to the initiation of MSCI, a biological process, occurring in P, that ensures transcriptional repression of sex chromosomes genes. MSCI is similar to the meiotic silencing of asynapsed chromatin (MSUC), a biological process that prevent transcriptional activation of any pair of autosomes not properly aligned to the aim of eliminate aberrant meiocytes bearing asynaptic autosomes [128,129,130]. Before MSCI starts, an important wave of histone acetylation (H4K5ac, H4K8ac and H3K9ac) preliminarily occurs on the X and Y chromosomes. From the stage of P onwards, concomitantly to MSCI, all histone PTMs involved in chromatin opening (H4K5ac, H4K8ac, H4K12ac, H3K9ac, H3K4me3) decrease except H4K16ac, which serves to H2AX phosphorylation expansion [131].

BRCA1, ATR and γH2AX are the most important factors involved in the inactivation of the sex chromosomes. Asynapsed chromosomal regions of sex chromosomes are detected by BRCA1, which recruits ATR kinase on the XY chromatin. Subsequently, ATR phosphorylates histone H2AX at S139 to give γH2AX which in turn links the mediator of DNA damage checkpoint 1 (MDC1) that amplifies γH2AX from the axes to the chromosome-wide domain and initiates MSCI. As a result of H2AX phosphorylation, many histone modification changes occur on XY body [128,132]. In mice, silent epigenetic marks involved in heterochromatin formation accumulate on the XY body and participate in MSCI process. In particular, H3K9me2 accumulates on sex chromosomes during the transition between the P to diplotene (D) stages, whereas H3K9me3 accumulates in the D stage [131,133]. Interestingly, the complete loss of bromodomain testis-specific protein (BRDT) results in a meiotic block of spermatogenesis that correlates to a reduction of H3K9me3 detection on XY chromosomes in P, negatively affecting the silencing of the X and Y chromosomes [134]. In addition, histone H2A ubiquitination is required for sex chromosomes gene inactivation in SPC. In particular, ubiquitination in K119 of H2A, catalyzed by ubiquitine-conjugating HR6B and ubiquitin ligase UBR2 enzymes, has been detected on chromatin regions subjected to transcriptional silencing of sex chromosomes genes [30,135]. 

Probably, H3K79me3, catalyzed by DOT1L, is involved in MSCI: Ontoso and colleagues showed that the sex body is enriched for H3K79me3 in P and D-SPC, contributing to the maintenance of repressive chromatin at the sex body [13]. 

In human, a similar histone pattern involved in MSCI of sex chromosomes is observed in P-SPC, mostly for H3K9me2, H3K9me3 and H4K20me3 [132]. 

SUMO-1 is strongly detected in the sex body in P-SPC, suggesting a possible involvement of SUMOylation in meiotic sex chromosome inactivation and XY body formation [136]. SUMO-1 distribution supports a possible role in constitutive heterochromatin formation during meiosis. Indeed, in P it colocalizes with heterochromatin mark H3K9me2 in the same areas [137]. Additionally, SUMO-1 may initiate this process localizing in the sex body before γH2AX accumulation [138]. In human SPC, SUMO-1 is largely detected on sex chromosome axes, “pseudo autosomal regions” and on X and Y centromeres playing a role in human MSCI. Conversely to the situation observed in mice, in human SPC SUMO-1 is no longer detected within the sex body and pericentromeric heterochromatin, but only at the centromere level [33]. 

A select set of sex-linked genes escapes silencing and it is reactivated in SPT through a cooperation of several histone PTMs. A cross-talk between H2A ubiquitination and H3 acetylation in post-meiotic SPT seems to play a key role in this phase. In fact, the loss of H2AK119ub, previously fixed in P-SPC in order to ensure a correct MSCI, promotes the accumulation of H3K27ac linked to active transcriptional activity [139]. Interestingly, H3K4me2 also has a role in the sex-linked genes escape. In detail, the establishment of H3K4me2 on sex chromosomes from meiosis to round SPT seems to be linked to the deposition of two histone modifications important for active gene expression: K-cr and H3K4me3 [135,140]. Histone K-cr is a novel PTM recently characterized and described to mark specifically the promoter of XY-linked genes. It confers resistance to transcriptional repressors and facilitates gene re-expression after MSCI [141]. Interestingly, for a positively silencing escape in post meiotic SPC, low levels of H3K27me3 are required on the same promoters marked by K-cr [35]. 

### 3.3. Spermiogenesis

During spermiogenesis, round SPT undergo massive morphological changes, critically important for fertility. Such a metamorphosis includes formation of the acrosome and tail, nuclear condensation and elongation, cytoplasm elimination. Nuclear condensation is strictly related to massive chromatin reorganization [142,143] which implicates structural and functional chromatin reprogramming [93]. Indeed, during germ cell progression from SPC to elongating SPT, the incorporation of histone variants and a new histone code organize quickly disassembling nucleosomes. These new chromatin domains, in agreement with transient DSBs and haploid expression of transition proteins (TNPs) and protamines (PRMs), facilitate histone displacement and subsequent histone-to-protamine exchange [93].

In mammals, histone displacement preserves a small fraction of chromatin (about 1–2% in mouse and 15% in human) condensed by histones. This shows a number of PTMs. As a consequence, SPZ contain differentially packaged chromatin domains: (i) the nucleo-protamine chromatin that represents the bulk of DNA in a highly-compact toroidal structures and (ii) the nucleo-histone chromatin that represents a small fraction of DNA in a more relaxed and transcriptionally flexible nucleosomal organization [93]. Recently, it has been demonstrated that the nucleo-histone chromatin contains hypomethylated DNA and a number of histone modifications.

Histone acetylation plays an important role in destabilization and remodeling of nucleosomes [144,145,146]. 

H4 acetylation shows a spatial distribution pattern during spermatogenesis: it is high in male stem cells and is removed during meiosis, but a H4 hyperacetylation observed in the early elongating SPT affects the nucleus in a global manner [97,147]. This distribution, then, changes during the elongation and condensation stages and finally disappears following an antero-caudal movement in condensing SPT, reinforcing the hypothesis of a direct link between histone acetylation, their replacement and nucleus condensation [97]. Noteworthy is that the re-acetylation of H4 on K5, 8, 12 and 16, in elongating SPT is a pre-requisite for histone-to-protamine exchange [97,110,148,149,150,151,152]. In detail, H4K5ac, H4K8ac and H4K12ac are highly expressed in elongating SPT and disappeared in condensing SPT [97,148], while H4K16ac can only be detected in elongating SPT [148]. Accordingly, in humans, H4 hyperacetylation acts as a preliminary event useful for histone displacement beginning during spermiogenesis (Figure 5A). In detail, H4K8ac and H4K16ac detected in round SPT, localize on a doughnut-like structure, implicated in the histone removal occurring concomitantly to acrosome and perinuclear theca development [153]. Probably, H4 hyperacetylation is preceded by H4 phosphorylation during SPT differentiation. Indeed, during spermiogenesis, the increase of H4 acetylation in elongated SPT occurs after H4S1ph decline in round SPT [154,155]. Interestingly, the pattern of TNP2 accumulation appears to correlate with H4S1ph reduction in round SPT, suggesting that H4S1ph may act during histone removal [121]. 

Although hyperacetylation of additional histones such as H3 has been found to co-exist with H4 hyperacetylation [97,156], less attention has been given to the potential involvement of these histone modifications in germ cell chromatin remodeling. In particular, H3 acetylation occurs during SPT elongation and nuclear condensation, with a strong increase of H3K9ac, H3K14ac, H3K18ac and H3K23ac in elongated SPT prior histone displacement [98]. 

Interestingly, multiple histone methylations have been identified in elongating SPT (H3K4me2, H3K4me3, H3K9me2, H3K9me3, H3K27me3, H3K79me2 and H3K79me3) [98,99,153,157]. 

Active histone modifications H3K4me1/me2/me3 increase in round SPT, preferentially associated with euchromatic regions in order to control active gene expression involved in this differentiation phase [99].

Particularly, H3K4me2, together with H4 hyperacetylation, contributes to chromatin opening, indispensable for histone-to-protamine exchange [151]. Accordingly, H3K4me2 is highest in the final steps of spermiogenesis when protamination and acrosome formation occur [99]. Furthermore, H3K4me3, specifically located in the elongating SPT nuclei, is recognized by C-terminal PHD finger of Pygopus homolog 2 (PYGO2), that recruits HAT to regulate H3 acetylation (Figure 5B). Accordingly, a reduction of PYGO2 induces a decrease of H3K9ac and H3K14ac levels, negatively affecting histone H3 removal, and influencing *TNP* and *PRM* gene expression, thus causing the abnormal nuclear condensation which leads to male sterility [158]. 

H3K4me3/me2 are also recognized by PHD Finger Protein 7 (PHF7), specifically located in the elongating SPT nuclei, through its PHD domain. Interestingly, PHF7 also catalyzes H2A ubiquitination by its RING domain, to promote histone removal. Accordingly, in *PHF7*-null SPT, the H2A ubiquitination is dramatically decreased, resulting in the histone retention and PRM replacement defect [34] (Figure 5C). 

H3K36me3 may regulate histone-to-protamine transition by activating *TNP* and *PRM* gene expression. In this context, the disruption of histone methyltransferase SET domain-containing 2 (*SETD2*) causes complete loss of H3K36me3 in SPC and SPT and arrests the development at round-SPT stage, impairing *TNP* and *PRM* gene activation, thereby resulting in sterility [16]. Contrarily, a targeted disruption of *JHDM2A*, a specific H3K9me2/1 demethylase-also known as KDM3A-results in a complete loss of TNP1 and PRM1 expression, defective chromatin condensation and infertility [18]. In fact, JHDM2A directly binds to and controls H3K9 methylation at the promoter of *TNP1* and *PRM1* genes, regulating sperm genome packaging and chromatin condensation [18].

Additionally, H3K79me3 is detected in rodent elongated SPT and precedes histone removal overlapping with H4 hyperacetylation, thus suggesting that both H4 hyperacetylation and H3K79 methylation characterize the histones shortly before they are largely replaced by TNPs and later by PRMs [157].

In elongating SPT, H4 hyperacetylation has also the function to weak the affinity to DNA in order to facilitate DSBs induction by DNA Topoisomerase II beta (TOP2β). This latter relaxes DNA and reorganizes chromatin loops [159]; its activity results in the activation of PARP1 and PARP2 [159] that covalently attach PAR to histone H1 and mediate its removal, an essential step for SPT chromatin remodeling. Accordingly, it has been shown that genetic deletion of *PARP1* gene causes a defective nuclear condensation, an increase of DSBs in sperm and an abnormal histone retention [32]. PARP activity is required for serine/threonine-protein kinase ATM/ATR pathway. The ATM signaling network is an important part of DNA repair pathways [160,161]: in fact, ATM and ATR interact to form H2A.X in response to DNA DSBs. MDC1 binds to γH2AX [162] and recently, in elongating SPT, a new protein kinase called TSSK6 has been identified to be responsible for the H2A.X phosphorylation during the histone-to-protamine transition (Figure 5D) [31]. Indeed, *TSSK6* knock-out male mice show SPZ with elevated levels of H3, H4 and the precursor and intermediate of PRM2 [31]. 

Interestingly, MDC1 recruits: (i) ubiquitin E3 ligase RNF8 to ubiquinate H2A and H2B and replaces them for H2AFZ/H2B in nucleosomes (Figure 5E), (ii) TIP60 (Tat-interactive protein or KAT5) which together with EPC (Enhancer of polycomb homolog 1) form mammalian nucleosome acetyltransferase complexes of H4 (NuA4) [24]. In detail, activity of this complex also acetylates H2AFZ to locally open chromatin structure [163] (Figure 5F).

Recently it has been shown that Nut, a testis-specific factor exclusively expressed in post-meiotic cells, interacts with CBP/p300 HAT to enhance acetylation of H4 at both K5 and K8 [164] (Figure 5F). Interestingly, acetylation of histone H4 on residues K5 and K8 provides a binding site for the BRDT. This is a testis specific BET protein that has two functional bromodomains: the first (BD1) requires both H4K5 and H4K8 acetylation for the binding, whereas the second (BD2) specifically binds to H3K18ac [7]. BRDT participates to histone eviction, histone proteasomal degradation and chromatin packaging [7,165]. In detail, BRDT binds the N-terminus of SMARCE1, a member of SWI/SNF family of ATP-dependent chromatin remodeling complex to facilitate histone removal [166]. Interestingly, in mice, the lack of *BRDT* results in male sterility due to th production of the morphologically abnormal SPT [167]. In fact, in the absence of BRDT-H4 binding, TNP2 and PRMs are unable to properly localize within the nuclei of elongating and condensing SPT and the histone replacement does not occur, resulting in altered chromatin condensation and infertility [168] (Figure 5G).

At the same time, histone ubiquitination dependent by RNF8 appears to promote H4K16 acetylation, by regulating the localization of HAT MOF on the chromatin. Thus, the MOF-mediated acetylation of H4K16 initiates global histone removal [140] (Figure 5G). Accordingly, *RNF8* deficiency results in the nucleosome removal abnormality during spermiogenesis, which causes infertility in male mice [27,140].

The dynamic balance between acetylation and deacetylation of histones is essential for a correct histone displacement. Recently, Bell and colleagues attribute to Sirtuin 1 (SIRT1), a member of the NAD^+^-dependent deacetylase, a role in H4 acetylation and histone-to-protamine transition. In fact, germ cell-specific *SIRT1* knock-out mice display reduced male fertility due to decreased SPZ number and an increase in SPZ with chromatin condensation defects [26,35]. These defects are related to a decrease of acetylation levels of H4K5, H4K8 and H4K12 in elongated SPT and altered TNP2 nuclear localization [26].

In agreement, the use of an HDAC inhibitor, trichostatin A, results in a significant reduction of the number of SPT and severe male infertility [169,170].

Interestingly, mice lacking *ZMYND15*, a testis-specific protein present in the nuclei of haploid SPT that interacts with HDAC, show defects in elongated and elongating SPT causing male sterility [171]. 

In SPT, simultaneously with hyperacetylation, a newly histone K modification has been identified as crotonylation [141]. This regulates testis-specific genes activation in post-meiotic germ cells [141]. Since the discovery of K-cr, several regulatory enzymes for histone K-cr have been described. In particular, Chromodomain Y-like transcription corepressor CDYL negatively regulates histone K-cr by acting as a crotonyl-CoA hydratase to convert crotonyl-CoA to β-hydroxybutyryl-CoA. CDYL-catalyzed downregulation of histone K-cr is intrinsically linked to its transcription repression activity and functionally implemented in spermatogenesis by influencing the reactivation of sex chromosome-linked genes in round SPT and histone replacement in elongating SPT. Interestingly, *CDYL*-deficient male mice show reduced fertility, decreased epididymal sperm count and sperm cell motility and dysregulated histone K-cr [35].

## 4. Histone PTMs in SPZ: Their Implication in Sperm Quality and Fertility in Human

Sperm histone retention is not random. It occurs at specific genomic regions [172,173]. In fact, retained histones enriching imprinted loci are important for embryonic development [37,174,175].

Post-translationally modified histone residuals in sperm chromatin act as a “tag”, destined to become a kind of “stable memory”, essential for the establishment of epigenetic information in the offspring [43,176,177]. 

Several histone PTMs have been identified in mouse and human sperm and most of them are shared between the two species. Recently, Luense and collaborators have been shown in mouse sperm 17/20 PTMs of histone H3, 10/12 PTMs of histone H4 and 5 PTMs of histone H2A/H2B shared with human sperm [36]. In detail, most histone PTMs common in the two species include a number of modifications such as H3K9me1/2/3, H3K23ac, H3K27me3, H3K36me1/2, H3K79me1/2, H4K5/8/12/16ac and H4K20me1/2/3. Interestingly, in mouse sperm, 26 PTMs on histone H1, H2AB, H3 and H4 have been identified. Four of them (detected on histone H2B and H3) are not described in any tissueor cell line, suggesting that some histone PTMs may be specific for SPZ [151]. Furthermore, a potential crosstalk between histone PTMs has been observed in mouse sperm. For example, H3K27me1 may influence H3K36me1 deposition. In fact, this last only occurs where H3K27 is also methylated [178]. Consistent with histone PTMs detected in mouse sperm [178], high levels of H3K4me1, H3K9me1, H3K9me2, H3K27me3 and H4ac have been observed also in human sperm [179]. In particular, the study of subcellular localization of histone PTMs in human sperm surprisingly shows that H4ac and H3K27me3 are present not only in the nucleus of sperm, but also in the mid-piece of the tails [179]. In this regard, residual histones in the mitochondria-containing mid-piece have been found: this observation may suggest H4ac and H3K27me3 presence in sperm tail, even though mitochondrial DNA is histone-free [180]. Undoubtedly, further studies are needed to clarify this observation.

Interestingly, a correct evaluation of sperm epigenetic quality not only depends on a correct PRM ratio (PRM1/PRM2), but also on a specific histone profile correlated with protamine levels [179]. In fact, a strong correlation between PRM2 and H3K9me2 levels is observed in human sperm head, whereas low PRM2 levels are correlated with low levels of H3K9ac, H3K9me1, H3K27me3, H3K36me3 and H3K79me1.

In this context, the changes in histone retention and histone modifications may have important role in sperm function, sperm quality and embryo development [37,174,181]. Defects in either the replacement or the modification of histones might result in azoospermia (total absence of SPZ), oligozoospermia (low sperm concentration) or teratozoospermia (sperm with abnormal morphology), leading to male infertility. Furthermore, the conservation in sperm epigenomic packaging suggests a potential role for paternal histones in early developing embryos [37,174,182]. Thus, epigenetic abnormalities in the genome of infertile men may cause post-fertilization errors [183,184,185,186]. 

A growing body of evidence suggests a role for different histone tail modifications in male infertility, particularly affecting sperm function at many levels. There are, however, few data specifically concerning the role of histone tail modifications in the setting of human infertility, and the available studies are, in general, limited to a handful of specific modifications, specifically acetylation and methylation on K residues of H3 and H4 histones. 

The site-specific-K acetylation of histones is known to regulate genomic histone retention [186] and transfer of epigenetic information to the oocyte [187]. H3K9ac is present in ejaculated SPZ. In sperm from fertile men, it is associated with specific regions of the sperm genome (i.e., promoters, exons and intergenic regions), some of which are involved in embryo development, defining an epigenetic code that could be transmitted to the embryo and influence gene expression after fertilization. This distribution pattern of H3K9ac becomes modified in sperm from oligozoospermic infertile men (men with idiopathic infertility and aberrant sperm chromatin condensation after Intra Cytoplasmatic Sperm Injection (ICSI), no pregnancy resulting or pregnancy ended in abortion) [186]. 

In addition, decreased acetylation of H4 has been described in men with impaired spermatogenesis [188]. Recently, Schon et al. found a significant decrease in overall acetylation on H4 (aa 4–17) when they compared sperm samples from asthenoteratozoospermic with normospermic men [189]. These recent data are in accordance with observations that the levels of histone H4 acetylation (specifically acetylated at K5, 8, 12 and 16) are significantly lower in ejaculated SPZ of infertile men exhibiting round SPT maturation arrest [190]. 

Additionally, acetylation of histone H4 at K12 (H4K12ac) is observed prior to full decondensation of sperm chromatin after fertilization, suggesting an important role for the regulation of gene expression in early embryogenesis. During fertilization, when the pronuclei are formed, the paternal pronucleus exhibits a strong acetylation signal on H4K12ac, while in the maternal pronucleus, there is a permanent increase of H4K12ac until pronuclei fusion. In this context, aberrant H4K12 acetylation in selected developmentally important promoters in the sperm of subfertile men reflects an insufficient sperm chromatin packaging, affecting the transfer of epigenetic marks to the oocyte [187]. Thus, H4K12ac could represent a potential factor for epigenetic mediated infertility. Interestingly, sperm HAT activity is found to be positively correlated with the sperm DNA fragmentation index (DFI) in normozoospermic men [191]. Recently, Rajabi et al. show no relationship between the H4K12ac level and DFI in male pronuclei derived from SPZ with DNA fragmentation less than 30%. When DNA fragmentation is more than 30%, there is an inverse relationship between DFI and level of H4K12ac in male pronuclei. These findings demonstrate that increased level of damage results in a decreased level of sperm H4K12 acetylation, because DNA strand breaks are likely to cause inappropriate localization of acetylated histones in male chromatin [192].

Recently, H3K4me2 epigenetic mark has been used for the identification of SPZ with decrease quality and immature chromatin [193]. In this study, Štiavnická and colleagues analyze semen samples of a large cohort of men classified by pathological semen quality. In particular, H3K4me2 levels correlate with sperm immaturity (high DNA stainability, % HDS) and are significantly higher in asthenozoospermic (with reduced sperm motility) and oligoasthenozoospermic (with reduced sperm concentration and motility) than in normospermic samples. Interestingly, H3K4me2 has been detected at the promoters of transcriptionally active genes crucial for spermatogenesis [182]. Sperm alterations in H3K4me2 pattern impair the development and survival of the offspring and are paternally inherited across generations [194]. Paternal H3K4me2 is also involved in the regulation of gene expression during early embryonic development [195,196]. Thus defective levels of paternal H3K4me2 could be responsible for aberrant early embryogenesis [195,196].

In sperm histone code, another molecular marker associated with sperm incompetence and decreased fertility is represented by H3K9 methylation. Interestingly, asthenoteratozoospermic (exhibiting severe abnormalities in total and progressive motility and in morphology) and asthenozoospermic (exhibiting less severe abnormalities in total and progressive motility) samples show altered H3K9 methylation when compared to normospermic patients [189]. H3K9 methylation is also considered hallmarks of heterochromatin: the combination of H4K20 and H3K9 methylation leads to a highly condensed and transcriptionally repressed state [197]. Recent evidence suggests that H3K9me3, in association with H4K20me3, from paternal heterochromatin is transmitted from sperm to embryo [198], highlighting the importance of this specific histone PTM on fertility and embryogenesis.

H3K36me3 (together with H3K4me1, H3K9me2, H3K4me3 and H3K79me2 marks) is reported in poorly functional human sperm [199], consistently with those observed in mouse sperm [178].

In a recent study, genome-wide analysis on sperm chromatin regions from patients with poor embryogenesis after IVF and men with altered protaminated sperm, reveals that five of the seven infertile men showed non-programmatic retention of histones. Histone modifications show a reduction of developmental promoters enriched with H3K4me3 and H3K27me3 in most infertile men, although the localization of modified histones appears unaltered [200]. 

## 5. Sperm Histone PTMs: A Potential Role in Embryo Development

After fertilization, in male pronucleus, the paternal genome undergoes a significant transformation: (i) PRMs are replaced by maternal histones; (ii) genome-wide demethylation occurs before zygotic genome activation; (iii) a strong de novo methylation fixes a new methylation pattern [201,202]. This epigenetic reprogramming in early stages of embryogenesis ensures totipotency, pluripotency acquisition and a correct initiation of embryonic gene expression. Many histone PTMs detected on sperm genomic regions remain associated with the paternal genome during de novo chromatin formation that occurs after fertilization [202,203,204,205], thus sperm histones may play a key role for a proper embryonic reprogramming and transgenerational epigenetic inheritance [37,204,206,207,208].

As described above, the distribution of sperm derived histone PTMs on zygotic genome is not random, but follows a specific pattern [209,210]. In detail, sperm retained histones are enriched at development-related loci, including imprinted gene clusters, binding sites of chromatin insulator protein CCCTC-binding factor (CTCF) and promoters of developmental transcription and signaling factors [37,174,182,187,206,211,212]. Most of them have unmethylated CpG-rich promoters and are enriched of H3K4me2/3 or H3K27me3 [111,174,182,194,205,206,209,212,213]. Actually, H3K4me3 marks promoters of microRNA clusters, transcription and signaling factors (i.e., STAT3, KLF4/5, SOX7/9), developmental genes and paternally expressed imprinted genes (i.e., *MEST*, *BEGAIN*, *DLK1*, *RTL*). In contrast, H3K27me3 marks promoters of several pluripotency factors (i.e., *SOX2*, *OCT4*) and *HOX* cluster genes [37,206,209,212]. These histone PTMs may protect unmethylated CpG-rich promoters from de novo methylation occurring in early embryo development to ensure transcriptionally activation [206,212]. H3K4me3 and H3K27me3 can also co-localize on promoters of developmental regulatory loci in mature SPZ forming a bivalent signature linked to transcriptionally repression. H3K27me3 removing from bivalent signature after fertilization induces gene expression activation. Interestingly, genomic regions with this bivalent signature are similarly marked in embryonic stem cells [37,214].

It is known that heterochromatin markers H3K9me3, H4K20me3 and H3K64me3 present in human sperm are transmitted to the embryo [198]. In particular, after fertilization, paternal H3K9me3 is recognized by maternal heterochromatin protein 1 (HP1) to propagate this mark over embryonic cleavage divisions. Thus, an intergenerational pattern of constitutive heterochromatin (cHC) in early human embryogenesis is determined by the paternal genome [198,203,209,215,216,217]. In mouse SPZ, H3K9me3 retained at satellite repeats, does not seem to be maintained in paternal pronuclei in the late zygotic stage after fertilization. Indeed, mouse embryo seems to re-establish paternal cHC via maternally inherited [198,202,204,209,216,217,218,219]. In addition, H3K64me3 and H4K20me3 are not found on mouse paternal chromatin in early stages of embryogenesis, whereas they are detected as late heterochromatin markers until compaction [220,221]. 

H4K8ac and H4K12ac have been detected prior to full decondensation of mouse sperm nucleus, suggesting that these epigenetic marks are transmitted by SPZ and persist in the zygote [187,203]. In human sperm, H4K12ac enriches the promoters of genes involved in: (i) embryonic development; (ii) reproductive processes; (iii) chromatin modification [187]. The same role is played by H3K9ac that, in human SPZ, is associated with promoters and interacting exons involved in: (i) metabolic process; (ii) neuronal and brain development; (iii) signal transduction [186]. Interestingly, in human sperm H4K12ac and H3K9ac are also found as bivalent signature on gene promoters involved in reproductive processes and embryonic development [187]. 

Although studies to improve molecular and embryonic understanding of histones in both human and mouse SPZ are still in progress, their detection adds a higher value to sperm chromatin features. A large set of gene loci are significantly marked by sperm histones, ensuring proper totipotency and developmental and imprinted gene regulation as well as appropriate chromatin rearrangement in early embryo. Overall, sperm histone PTMs may act as “bookmarks” to propagate the epigenetic profile of the paternal genome during early stages of embryonic reprogramming.

## 6. What Is Known about CircRNAs in the Reproductive Field? 

CircRNAs are ancient ncRNAs with recently discovered regulating functions. Thought to be junk by-products in gene transcription, accumulating studies have been then performed to explore circRNA expression profile in several cell types and diseases, as well as in various stages of cell differentiation. Interestingly, circRNAs have a multi-faceted biological function and different localizations in cell. Furthermore, their regulating role is not isolated, but trapped in a complicated network involving mRNAs, miRNAs and proteins. The advancement of RNA-seq technologies, together with the development of computational pipelines, has led to an explosion in circRNA research in several directions. To date, the role of circRNAs in the reproductive field has been poorly investigated.

### 6.1. The Making of a CircRNA: Biogenesis, Subcellular and Tissue Localization

CircRNAs especially derive from protein-coding exons that are covalently arranged in a non-canonical order. *Backsplicing* mechanism has been suggested for circRNA biogenesis [222]. In the canonical linear splicing, an up-stream (5′) splice donor site is linked to a down-stream (3′) splice acceptor site. By contrast, on the basis of the backsplicing, a down-stream (3′) splice donor site is covalently linked to an up-stream (5′) splice acceptor site [223]. Such an event may occur both co- and post-transcriptionally [224]. Canonical spliceosomal machinery may be employed in linear as well as in backsplicing, although how exactly this happens still remains incompletely understood. Accordingly, the pharmacological inhibition of the spliceosome in rat primary hippocampal neurons causes a significant shift in circRNA landscape [225]. With the development of deep sequencing, several types of circRNAs have been discovered and classified into five subsets: exonic, intronic (ciRNAs), exon-intron (EIciRNAs), antisense and sense overlapping. The exonic ones, the most abundant in cells, are preferentially located in the cytoplasm; ciRNAs and EIciRNAs are primarily located in the nucleus where they may regulate the transcription of their parental genes. In addition, “antisense” are circRNAs having their gene locus overlapped with the linear RNAs, but transcribed from the opposite strand; “sense overlapping” are circRNAs transcribed from the same gene locus as the linear counterpart, but not classified as exonic or intronic; “intergenic” represent circRNAs located outside known gene locus [226].

Mutation experiments suggest that the efficiency in circRNA biogenesis is positively influenced by RNA polymerase II integrity and elongation velocity [227]. Additionally, inverted repeats and/or inverted ALU repeat elements (IAREs) in the introns flanking the exons bringing the splice sites contribute to RNA circularization, as cis-elements [228]. Trans-factors have also been reported to promote circRNA biogenesis. In this regard, RNA binding proteins (RBPs) are likely candidates for such a process. Their role strongly varies depending on the circRNA, tissue type, cell type and biological circumstances. Quaking (QKI) positively regulates circRNA biogenesis during the epithelial-to-mesenchymal transition [229]; muscleblind (MBL) specifically regulates the expression of circMbl, but not other circRNAs in *Drosophila*, mouse and human [230]. Both RBPs may dimerize to facilitate circularization [231]. By binding to specific exon-intron junctions, RNA-binding protein FUS (FUS) protein may also regulate circRNA formation, especially during embryonic stem cell differentiation [232]. Once synthesized, circRNAs accumulate in cells and are strongly in competition with linear transcripts. The biogenesis of circRNAs has, in fact, been suggested to have a negative effect on linear mRNA production [233], thus envisaging that circRNA generation may be an active process to modulate gene expression and protein synthesis. 

Whether there are proteins/mechanisms that facilitate circRNA formation, there should also exist a way to regulate circRNA degradation in cells. Interestingly, RNA editing, especially A-to-I signature by the deaminase ADAR1, destabilizes the base-pairing to further suppress circRNA biogenesis in human and mouse cell culture [234]. Another mechanism that limits circRNA formation consists of the disruption of ALU inverted repeats-dependent structures by the RNA helicase DHX9 [235]. 

Since their covalently closed structure, lacking poly-A tails and 5′ caps, circRNAs can escape de-adenylation and de-capping, thus they do not be accessible to several canonical RNA decay pathways. In fact, circRNAs are highly resistant to RNase R, an exonuclease that digests almost all linear RNAs [236]. In this way, circRNAs can accumulate in cells in a temporally regulated manner, owing to their high stability. However, as single-stranded RNAs with unpaired residues, circRNA may be subjected to degradation by other RNases, like RNase A. At moment, little information exists about mechanisms of circRNA degradation, especially in vivo. Recently, it has been demonstrated that *N*^6^-methylation of adenosine (m^6^A) recruits endonucleases potentially able to degrade circRNAs [237]. Extracellular vesicles, like exosomes, may pack up circRNAs protecting them during their circulation in blood or other bodily fluids. They may also constitute an alternative way to circRNA clearance, lowering their levels in cells by exocytosis [238]. Intriguingly, the vesicular transport of circRNAs in body potentially suggests that these RNAs may also function as signaling molecules. Whether circRNAs may regulate gene expression in distant cells other than where they are produced remains to be determined. Nevertheless, the existence of circulating circRNAs points to them as promising diagnostic biomarkers.

However, molecular mechanisms involved in biosynthesis, modulation and degradation of circRNAs are still largely unknown. A possible impact of epigenetic changes on circRNA biogenesis has never systematically investigated. In this regard, it has been demonstrated that knockdown of *DNMT3b*, which encodes a DNA methyltransferase, shows a global change in circRNA pattern, independently by the expression of the corresponding linear transcripts, suggesting a crosstalk between gene methylation and circRNA biogenesis [239].

What may be the biological function of circRNAs is still under investigation. 

One of the most acclaimed roles is the modulation of miRNAs, as their selective sponges [240]. CircRNAs interfere with the binding procedure between miRNAs and their target mRNAs, consequently weakening the repressive effects of miRNAs on mRNA translation. Thus, circRNAs are part of the competitive endogenous RNA (ceRNA) network (ceRNET). An essential prerequisite for a circRNA to be a ceRNA candidate is that to contain multiple binding sites for miRNAs. One of the better examples of this sponging function is given by CDR1as that has 73 seed-binding sites for miR-7, thus binding to miR-7 about 10 times higher than any other linear transcripts. Together with CDR1as, circSry is also a nice candidate for sponging activity by binding to miR-138 in 16 conserved sites [241]. A similar sponging activity may also explain the relationship between circRNAs and proteins; one of the first examples came from a study concerning circMbl in *Drosophila*. The excess of linear transcript and, therefore, of protein MBNL1 promotes the biogenesis of the circRNA that—by tethering both linear transcript and protein MBNL1—induces an autoregulatory circuit [230]. Starting from this evidence, circRNAs that bind to RBPs, can act as decoys/transporters for these factors or serve as a protein scaffold in order to facilitate the accumulation, the recruitment and/or the colocalization of enzymes and their substrates in a cellular district [242,243]. Such a cooperative action may create in the cytoplasm a molecular reservoir of proteins in order to facilitate a prompt response to extracellular stimuli. With a similar mechanism, circRNAs may act as mRNA traps. Since most circRNAs consist of exons which may also be involved in mRNA biogenesis, it is plausible a competition between circRNA formation and linear mRNA biogenesis exists [244].

In the nucleus, where ciRNAs and EIciRNAs found their preferential subcellular localization, these molecules may be involved in the transcriptional regulation of their parental genes [245]. On the other hand, since most circRNAs are exon derivatives and are located in the cytoplasm where translation takes place, it is not surprising that circRNAs may be translated into a protein. However, this possibility has been for a long time considered remote due to circRNA structure devoid of 5′cap and poly(A) tail, essential elements for cap-dependent translation. Further studies have revealed that circRNA translation may occur through internal ribosome entry site (IRES) [246]. *N*^6^-methylation of adenosine is also a driving event for translation initiation [247]. What may be the biological function of circRNA-derived peptides is still an open question. These peptides are often truncated version of the canonical proteins lacking essential functional domains. Therefore, they may act as decoys or modulators of alternative protein complexes [248]. More intriguing is the possibility that circRNA-derived peptides may be expressed just in stress conditions or function in different cellular compartments than canonical proteins and several other modulating factors of circRNA translation may exist in cells (Figure 6 summarizes circRNA biogenesis and functions).

Spatial distribution of circRNAs in cells is still under investigation, with several difficulties due to low expression levels as well as sequence similarity of circRNAs compared to their linear mRNA counterparts. An elegant experimental approach consisting in a padlock technology in combination with in situ sequencing has been proved to be useful for the identification of spatio-temporal expression of circRNAs at the subcellular level in cells and human tissues [249]. What is proven is that circRNAs are synthesized in the nucleus. ciRNAs and EIciRNAs stay there to interact with the splicing machinery in order to control transcription [245], whereas the preferential localization of most circRNAs is the cytoplasm, where they especially tether miRNAs and/or RBPs interfering with their pathways or they may be translated into proteins. On this basis, circRNAs need to be transported across the nuclear membrane. Nuclear export machinery has been found to sort circRNAs selectively on the basis of their length, both in *Drosophila* and human cells. Such a transport is under the control of two RNA helicases, URH49 and UAP56, that preferentially export shorter and >1200 nt circRNAs, respectively [250,251]. 

What may be the molecular mechanism guiding circRNA vesicular uptake/packaging is still unknown.

CircRNAs show distinct tissue- and cell-type specific expression [222,240]. The brain is highly enriched in circRNAs, with numbers equal to 65,731, whose expression specifically changes in different brain regions and is independent of the linear isoforms [90]. CircRNAs strongly accumulate in the brain with age and especially localize in synapses, suggesting their involvement in age-related brain diseases and in synaptic plasticity, respectively [252]. Interestingly, in comparison to other organs, human testis has numbers of circRNAs only secondary to that of brain [253]. However, circRNA saga in testis has more ancient origins.

### 6.2. CircRNAs: From Testis to Embryo

Back in 1979 was the first discovery of circRNAs in the cytoplasm of eukaryotic cell lines by electron microscopy [88], but these molecules were considered for a long time “junk” products of aberrant splicing. Several years later, in 1993, there was the first demonstration that a circRNA may have a role, just in testis, thanks to the discovery of a testis-specific circRNA from *Sex-determining Region Y* (*SRY*) gene [254]. At that time, many questions as for example, “Do circular transcripts exist for other genes?” or “What mechanisms achieve these molecules?” remained unsolved. Only in 2016 a complete payload of circRNAs—15,996—in testis by using next generation sequencing [253]. The large number of circRNAs in human testis strongly suggests the need of massive gene expression regulation during spermatogenesis. Most testicular circRNAs derives from exonic regions of genes and they are widely scattered on all chromosomes, including mitochondrial genome, in accordance to their host gene location. Alternative splicing, a very common phenomenon in eukaryotes, has also been observed for testis-derived circRNAs. Gene ontology (GO) analysis reveals that testicular circRNAs are closely related to germ cell progression, meiotic events, histone displacement, sperm motility and fertilization.

Mammalian spermatogenesis is a complex and finely regulated progression of germ cells with morphological distinguishable characteristics [255,256,257]. Hence, the idea to systematically identify the complicate transcriptome of each cell type. Interestingly, 15,101 circRNAs have been detected in mouse spermatogenic cells [258], with a dynamic pattern: 5573, 5596, 6689, 4677 and 7220 circRNAs have been found in spermatogonial germ cells, primitive type A SPG, PL SPC, P SPC and round SPT, respectively. The large number, especially in round SPT, allows to hypothesize an important control of circRNAs in SPT maturation [258]. Such an evidence has been strongly supported by the analysis of Rat Body Map transcriptome [259]. CircRNA repertoire has been scanned in 11 rat tissues and 4 developmental stages demonstrating that (i) circRNAs are evolutionary more conserved than linear mRNAs; (ii) circRNAs have higher tissue-specificity than mRNAs; (iii) in testis, circRNA expression dynamically changes depending on the age: it increases with sexual maturity and decreases with ageing. In accordance with evidence in mice, a high degree of circRNAs has been associated in rats to differentiation events in spermatogenesis, especially linked to flagellum morphogenesis and SPT development. However, circRNA biogenesis in tissues has been suggested as essential for that tissue performance, thus brain-specific circRNAs have been associated to “chemical synaptic transmission” as well as in testis they have been associated to “sperm motility acquisition” and “spermatid development”. Additionally, as brain, testis seems to be a very sensitive organ with age-dependent changes in circRNA levels, thus to suggest that these molecules may be harnessed as biomarkers for reproductive ageing [259]. Together with intratesticular production of circRNAs, testis-derived circRNAs stably exist in seminal plasma, probably in the form of protein-complexes, thus to strongly suggest their application as novel non-invasive biomarkers for male fertility [253].

The idea that circRNAs may be heritable molecules has directed scientists to their identification in embryos. One of the first report has developed a single cell RNA-seq transcriptome analysis technology to dissect the transcriptome of individual cells [260]. Abundant circRNAs—2891—have been profiled in mouse preimplantation embryos, part of them already expressed in mature oocytes, but with dynamically increased expression until the four-to eight-cell stage. This increased expression has been explained through the high degree of circRNA resistance to the global degradation that, instead, affects maternal linear mRNAs during the maternal to zygotic transition. Mouse embryonic circRNAs are potentially involved in chromosome organization, cell cycle regulation and DNA repair [260]. In order to create a favorable environment for embryo implantation, a complex crosstalk should exist between embryo and uterus. In mice, the day 5 is critical because blastocysts start the implantation process in the endometrium. To better understand a role of circRNAs in such a crucial developmental window, a complete profiling has been carried out in the endometrial tissues of early pregnant mice on day 5, comparing the expression levels of circRNAs between implantation and interimplantation sites [261]. This analysis strongly suggests that circRNAs may be related to endometrial receptivity and—by working as miRNA sponges—they may target several genes involved in embryo implantation. As in mice, a complete landscape has been traced in human preimplantation embryos [262]. Individual cells at seven consecutive stages (mature oocytes, zygotes, 2-cell, 4-cell and 8-cell embryos, morulae and blastocysts) have been sequenced and 10,032 exonic circRNAs from 2974 hosting genes have been annotated. Most circRNAs here identified are transcribed from maternal genes, therefore already present before fertilization. As previously discussed in mice, circRNA resistance to maternal mRNA degradation guarantees their persistence in preimplantation embryos [262]. 

What emerges from these studies is surely the maternal contribution in circRNA embryonic inheritance and that circRNAs are actively produced in both oocytes and ovary [263,264]. A genome-wide identification of human ovary circRNAs has been carried out showing circRNA dynamical regulation during ovarian aging, as already suggested for brain [90,263]. In the ovary, circRNAs have been associated with metabolic processes, serine/threonine metabolism and especially steroid hormone biosynthesis, a critical signaling for normal ovarian functionality. Similarly to the ovary, a circRNA profiling has been examined in both porcine oocyte and cumulus cells revealing 637 and 7067 circRNAs, respectively [264], with potential functions mainly in signal transduction and tight junction regulation, key events in the oocyte-cumulus cell crosstalk. 

All these profiling analyses pave the way to decipher functional significance of circRNAs during mammalian early embryonic development. Recently, a maternally expressed circRNA—circARMC4—has been selected to explore its role in oocyte maturation. Interestingly, circARMC4 knockdown significantly reduces the rate of oocyte maturation through chromosome misalignment and impairs early embryo development [264].

What is noteworthy is that most literature concerning the contribution of circRNAs in embryo development especially points to maternally expressed circRNAs, excluding that mature SPZ may be a potential circRNA source.

### 6.3. Sperm-Derived CircRNAs: Potential Modulators of Sperm Quality, a Special Focus

Despite that circRNAs have been identified in both mammalian testis and seminal plasma [253,258,259] and several lines of evidence have pointed to establish a potential role for these molecules in mammalian embryo development, a missing puzzle piece concerns a possible circRNA landscape in mature SPZ. Very recently, our group—for the first time—has drawn a complete payload of circRNAs in human SPZ, analyzing all the characteristics concerning these molecules by using bioinformatic approaches [265]. A total of 10,726 circRNAs have been identified, most of them are exonic, as already demonstrated in testis [253]. In accordance to their host gene localization, SPZ-derived circRNAs are widely distributed across all chromosomes, mitochondrial genome included. 

As previously discussed, morphological parameters are classically examined to identify high quality SPZ, since ejaculates contain a multitude of sperm cells and the good choice of a single sperm cell is decisive, especially in ICSI strategy. A great effort has been, also, focused on SPZ DNA integrity as an additional parameter to evaluate, especially perturbed by reactive oxygen species (ROS) molecules, considering that SPZ with a damaged DNA, still retain their ability to fertilize, but with serious implications for offspring health [266]. Therefore, it appears evident that a correct evaluation of sperm quality should consider a number of aspects, including morphological, genetic and epigenetic ones, and does not limit itself to an exclusive morphological exam.

With this in mind, a possible correlation between circRNAs and sperm quality has been addressed by analyzing circRNA pattern of expression in two different SPZ populations, isolated from semen samples collected from normospermic volunteers, distinguished on the basis of morphological parameters [265]. What appears evident is the existence of a cohort of differentially expressed circRNAs in SPZ having good and poor quality, respectively, that could be used as potential biomarkers. Most of them have been identified and selected as SPZ specific circRNAs, on the basis of a comparison between SPZ-derived circRNA and Dang et al. dataset [262]. Our interest, in fact, has been focused on circRNAs surely expressed in SPZ and in some human embryonic developmental stages, but not in human oocytes as potential epigenetic carriers of paternal origin. To confirm this, KEGG analysis has suggested an involvement of SPZ-derived circRNAs in important biological processes that could be linked to the first stages of embryo development such as DNA replication and cell cycle. Oocyte meiosis has also been raised as involving SPZ-derived circRNAs. 

However, whether paternal circRNAs are transferred into oocyte during fertilization, they undoubtedly should localize in sperm head [267]. With this in mind, circRNAs localization has been accurately checked in SPZ head- and tail-preparations in order to discriminate subcellular localization of the selected circRNAs. Therefore, differentially expressed circRNAs in good and poor quality SPZ have been carefully analyzed in order to distinguish SPZ-derived circRNAs as molecules of more interest because potentially involved in embryo development [265].

These data strongly encourage to include circRNAs in the pool of molecules able to influence sperm quality. Certainly, male fertility is in danger because of environmental pollutants, inadequate diet and lifestyles [268,269,270,271,272]. In addition, several lifestyle pastimes such as alcohol, tobacco and marijuana have been shown to have further negative effects on male reproduction [270]. In this regard, a huge literature focuses on endocannabinoid system as a key player in the multifaceted process of male reproduction. Such a system has been deeply characterized in vertebrates [273,274,275,276,277,278,279]. One of the main actors of the endocannabinoid system is the enzyme NAPEPLD, involved in the biosynthesis of the endocannabinoid anandamide (AEA) [280]. During early pregnancy, AEA levels are spatio-temporally regulated in the uterus, mediating reciprocal interaction with blastocysts, with lower levels in the receptive uterus and at the implantation site [281,282]. Interestingly, human and murine SPZ retain two circular isoforms of NAPEPLD (that we named circNAPEPLDiso1 and circNAPEPLDiso2). Both isoforms are expressed at low levels in testis and heterogeneously fluctuate in sperm samples analyzed suggesting that, as for genetic traits, an individually variable transmission of epigenetic traits may exist as well [283]. Otherwise than previously reported, circNAPEPLDiso1 and circNAPEPLDiso2 expression appears to be independent from the linear counterparts, although there is evidence demonstrating a competition between circRNAs and linear transcripts [230,233]. In order to suggest that circNAPEPLD may be a paternal cytoplasmic contribution to the zygote, circNAPEPLDiso1 and circNAPEPLDiso2 expression has been examined in murine-unfertilized oocytes, revealing low and high levels of circNAPEPLDiso1 and circNAPEPLDiso2, respectively. After fertilization, such a profile of expression, intriguingly, changes just for circNAPEPLDiso1 that may be directly involved in zygote development. Starting from this data, the attention has been especially focused on circNAPEPLDiso1 for investigations concerning miRNA and RBP sponging activity as well as its potential translatability. 

In line with results demonstrating the sponge activity of circRNAs towards miRNAs [240], circNAPEPLDiso1 has been suggested to physically interact with five miRNAs whose targets are potentially involved in several molecular mechanisms regulating the first stages of embryo development, most of them are known to be expressed in human oocytes, blastocysts and blastocoel fluid [284,285,286]. Our data strongly suggest that SPZ-derived circNAPEPLD may function as a decoy to inhibit the anti-proliferative activity of some oocyte-derived miRNAs, thus allowing cell proliferation, a massive event during the first stages of embryo development [283]. As strongly suggested for other circRNAs, circNAPEPLD also appears to be able to bind RBPs. Interestingly, 11 eIF4A3 binding sites have been found on circNAPEPLDiso1. Indeed, this RBP is frequently reported to be associated with circRNAs, especially at junction sequences, but its molecular function remains largely uncharacterized [287]. Additionally, the possibility that circNAPEPLDiso1 may be translated into a protein has also been considered. Such an analysis has suggested that circNAPEPLDiso1 is potentially translatable, with a predicted ORF of 352 aa, in both humans and mice. This protein has a shorter C-terminal compared with the protein encoded by the linear transcript (393 and 396 aa in humans and mice, respectively), but concerning the biological function of circRNA-encoded proteins it is still an intriguing open question [283].

Therefore, the idea that SPZ-derived circRNAs may join the repertoire of ncRNAs transferred from sperm to oocyte and mediate paternal transgenerational epigenetic inheritance is really exciting. However, research in this area is still poor.

### 6.4. SPZ-Dependent CeRNET Involved in Germ Cell Progression and Embryo Development

Since the understanding of circRNA role in paternal transgenerational epigenetic inheritance is at its infancy, much effort is required to shed light on circRNA contribution to embryo development and germ cell progression. The characterization of the circRNA landscape in human and mouse SPZ and the hypothesis of its possible involvement in the control of sperm quality suggest that circRNAs may be modulators of male fertility. Until today, no experimental evidence exists in support to this hypothesis. The only indication regarding the delivery of a circRNA from SPZ to the oocyte comes from Ragusa et al., 2019 [283]. What a paternal circRNA in the zygote may do is still a fascinating mystery. However, bioinformatic approach is a useful instrument to glimpse the molecular mechanism downstream a circRNA. Interestingly, several differentially expressed circRNAs identified in human SPZ of good and poor quality sequester miRNAs, inhibiting their repressive activity towards mRNA targets, known to be involved in both embryo development and germ cell progression. 

In order to avoid redundancy in the description of such molecular pathways, we will just describe a SPZ-dependent ceRNET for each biological process. To this end, two circRNAs have been carefully chosen from a list containing 148 differentially expressed circRNAs because they are up-regulated in SPZ of good quality [265]. Circ-RERE has been selected for the construction of a ceRNET involved in embryo development (Figure 7A). It harbors 5 miRNAs: Has-miR-550a-3p, Has-miR-335-3p, Has-miR-571, Has-miR-377-3p, Has-miR-105-5p. Interestingly, a huge literature argues that several mRNA targets work in important phases of embryo development: *SOCS* and *FOX* targets are spatio-temporally regulated in the endometrium, reflecting the acquisition of receptivity, maternal recognition of pregnancy and implantation [288,289]; *YY2* is one of the major regulators of mouse embryonic stem cell self-renewal and lineage commitment [290] as well as the RNA methyltransferases *NSUN4* is enriched in mouse tissues during embryo development [291]. Interestingly, *NANOS1* has an essential role in the central nervous system development [292]. Similarly, *SOD2* overxpression alleviates neural tube defects induced by maternal diabetes, suppressing cellular stress in diabetic embryopathy [293]. Conditional *PDGFRA* knockout embryos have skeletal and abdominal wall defects supporting the idea that this kinase regulates important steps of embryo development [294]. Since circ-RERE may be considered a marker of sperm quality, it is plausible that its deregulation in infertile men alters the expression level of key regulators of embryo development (Figure 7A).

Circ-NFIC has been selected among circRNAs up-regulated in SPZ of good quality [265] as a key molecule modulating downstream effectors involved in germ cell progression (Figure 7B). It harbors 5 miRNAs: Has-miR-612, Has-miR-92a-1-5p, Has-miR-890, Has-miR-181d-3p, Has-miR-92b-5p. Interestingly, several mRNA targets are known to be involved in germ cell self-renewal and differentiation: *MYC* and *CCR1* enhance spermatogonial stem cell self-renewal and maintenance [295,296]; *PLK1*, *RAD51*, *FOXJ2* and *SBF1* especially control meiotic events [297,298,299,300,301]; *SP1* and *RFX1* are prominent in haploid germ cells [302,303]. Mice lacking *USP2* have severe male subfertility since their SPZ rapidly became immotile and do not fertilize eggs [304]. *WT1* is specifically expressed in Sertoli cells and critically controls blood-testis barrier [305].

Furthermore, preliminary data by using a bioinformatic approach clearly indicate a potential contribution of paternal circRNAs in both embryo development and germ cell progression, suggesting that it is worth investigating SPZ-derived circRNA fate. 

## 7. Conclusions

SPZ originate from a complex biological process of germ cell differentiation, including events of self-renewal, commitment, recombination, DNA-repair, differentiation, gene expression. Histone PTMs are involved in mechanisms related to such events, as well as they have been characterized in SPZ and demonstrated to be evolutionarily conserved in loci related to imprinting. Any interference with events related to sperm development may affect the molecular landscape of sperm histone PTMs with intergenerational and potentially transgenerational effects. In this contest, it is conceivable that sperm cargo of histone PTMs may be a useful sensor of disrupted spermatogenesis. Likely, aberrant sperm epigenome might be causative or resulting from a disturbed sperm development.

In the era of assisted reproductive technology (ART) widely used to treat human infertility, a method for distinguishing high-quality sperm samples, beyond classical semen parameters evaluation (sperm density, morphology, motility and DNA fragmentation), appears compelling. Ejaculates contain a multitude of sperm cells with variable epigenetic landscapes. Such an aspect is quite relevant in case of ICSI, a technique for which just a single sperm cell, mostly morphologically selected, has to be transferred into the oocyte. It is now noticeable that male fertility, successful reproduction, embryonic development, offspring’s health cannot be solely attributed to sperm DNA integrity. Rather, the identification of all the potential modulators and contributors to sperm epigenome is an appealing goal of modern research in the field of reproduction as well as in ART. In this scenario, a growing role may be assigned to circRNAs. Their identification in SPZ, their structural stability, their modulation as a function of sperm quality as well as their detection in seminal plasma make these molecules very fascinating in terms of potential regulators of sperm epigenome. Furthermore, by using mice as experimental model, the paternal contribution in circRNA cargo of the embryo through the delivery of circNAPEPLDiso1 to the oocyte has been demonstrated. However, the characterization of a complete circRNA profiling in infertile patients is still a missing piece that would shed light on altered epigenetic mechanisms at the basis of sterility.

All these data clearly reveal that SPZ contribution to embryo development is surely more than haploid genome. SPZ preserve their epigenetic landscape during maturation, plastically shape it during epididymal transit in relation to environmental impact, to ultimately deliver it into the oocyte. Furthermore, a fine analysis of histone PTM and circRNA patterns in SPZ might be a particularly relevant tool to support the mere morphological analysis in order to correctly diagnose or accelerate diagnosis of cases of male infertility.

## Figures and Tables

**Figure 1 jcm-09-00640-f001:**
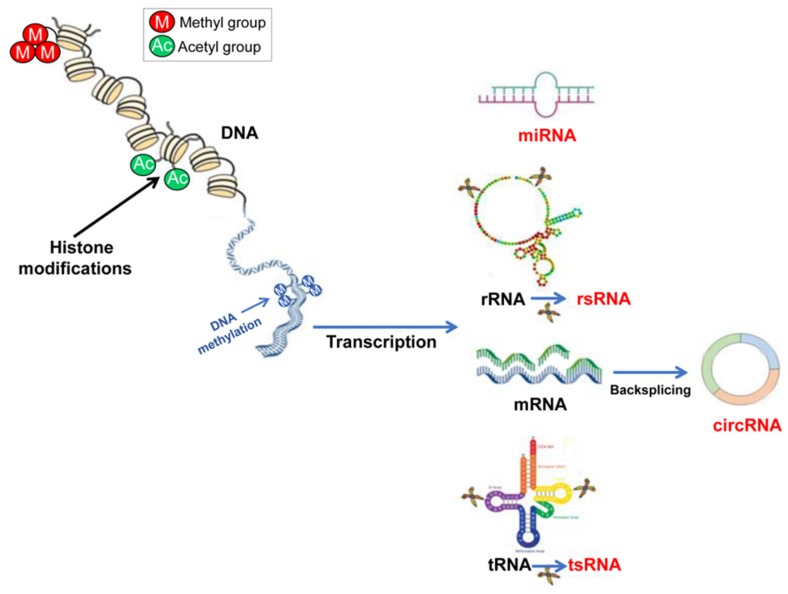
A schematic view of the main epigenetic processes. Gene expression can be epigenetically regulated at the transcriptional level, via DNA methylation and/or through histone histone PTMs able to conduct chromatin remodeling. The epigenetic regulation at the translational level is mainly under the control of ncRNAs. In sperm cells, recent results point to some classes of ncRNAs as mainly involved in their physiology, miRNAs, rsRNAs, tsRNAs and circRNAs, the last cited produced by mRNAs through a backsplicing reaction.

**Figure 2 jcm-09-00640-f002:**
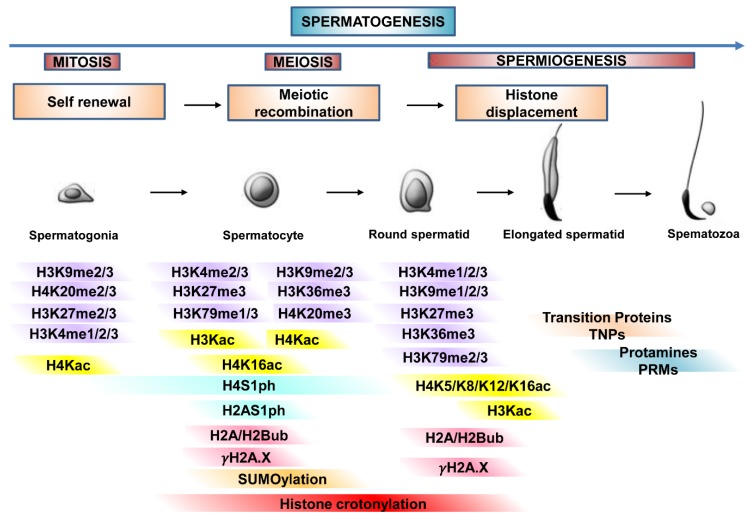
Histone PTMs regulate germ cell progression. The main histone PTMs involved in chromatin dynamic organization during proliferation, meiosis, and differentiation of germ cells, from spermatogonia to spermatozoa (spermatogenesis). In detail, histone PTMs during germ cell progression, self-renewal, meiotic recombination and histone-protamine transition.

**Figure 3 jcm-09-00640-f003:**
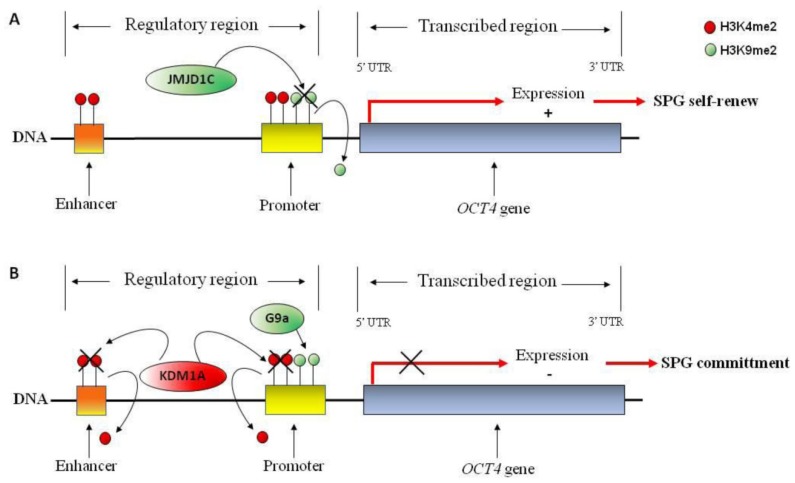
Hypothetical molecular mechanism involved in *OCT4* gene regulation in spermatogonia (SPG) stem cells. *OCT4* gene is represented. (**A**) Active transcriptional mark H3K4me2 (red circle) at the promoter and proximal enhancer of *OCT4* gene favors transcriptional activation; demethylation of repressive transcriptional mark H3K9me2 (green circle) by JMJD1C demethylase may participate to *OCT4* gene activation, favoring SPG self-renewal. (**B**) H3K9me2, by G9a methylase, blocks *OCT4* gene expression; H3K4me2 demethylation, by KDM1A demethylase, may participate to transcriptional repression of stem cell factor OCT4 favoring SPG commitment and differentiation.

**Figure 4 jcm-09-00640-f004:**
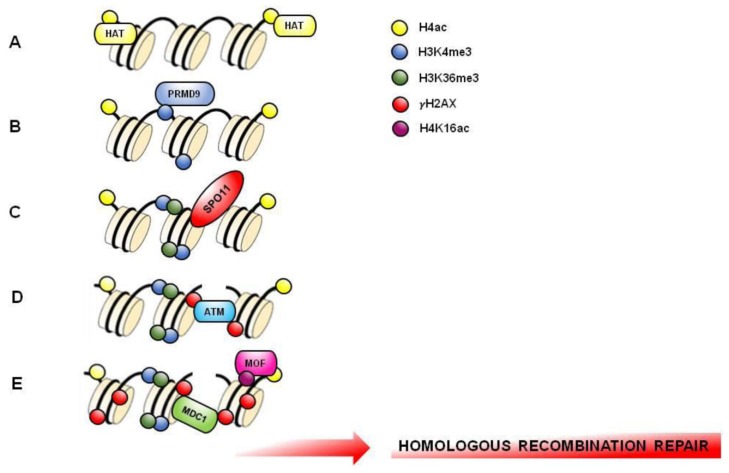
Hypothetical molecular mechanism involved in meiotic hotspots formation. DNA and several nucleosomes are represented. (**A**) Histone H4 acetylation (H4K5ac, H4K8ac, H4K12ac, H4K16ac) (yellow circle) is enriched in PL-SPC to favor an opened chromatin structure that facilitate recombination hotspots formation. (**B**) PRDM9 binds DNA and catalyzes H3K4me3 formation (blue circle) on hotspots. (**C**) H3K36me3 (green circle) associates with H3K4me3 on the same nucleosome to create a hotspot-specific signature recognized by endonuclease SPO11. (**D**) DNA double strand breaks (DSBs) formed by SPO11 triggers the phosphorylation of histone H2AX (γH2AX; red circle) by ATM kinase. (**E**) MOF acetylase induces H4K16ac (purple circle) facilitating MDC1 recruitment. MDC1 binds γH2AX and amplifies its phosphorylation to induce the recruitment of DNA repair factors starting homologous recombination.

**Figure 5 jcm-09-00640-f005:**
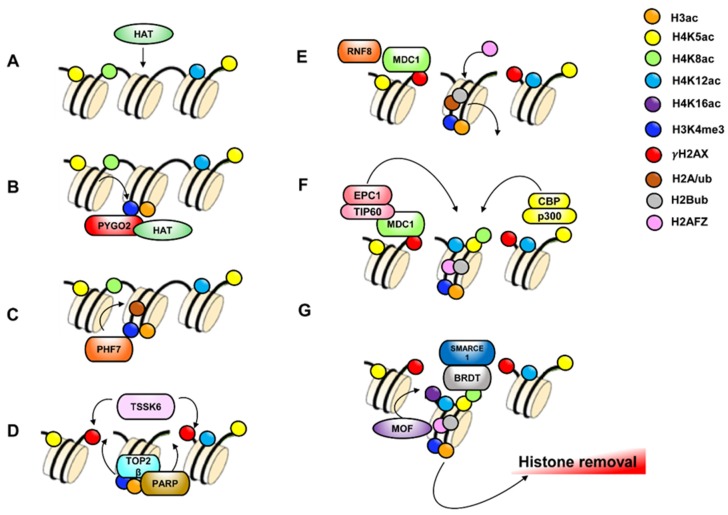
Hypothetical molecular mechanism involved in histone displacement. DNA and several nucleosomes are represented. (**A**) Histone H4 acetylation (green-yellow-light blue circles) catalyzed by Histone Acetyl Transferase (HAT) occurs to favor an opened chromatin structure that facilitate histone removal. (**B**) PYGO2 recognizes H3K4me3 (blue circle) and facilitate HAT recruitment to induce H3 acetylation (orange circle). (**C**) PHF7 recognizes H3K4me3 to induce H2Aub (brown circle). (**D**) TOP2β nuclease introduces DSBs and induces poly ADP-ribose polymerases (PARPs) activation. PARPs activity is required for TSSK6, a serine-protein kinase, that forms γH2AX. (**E**) MDC1 binds γH2AX and recruits ubiquitin E3 ligase RNF8 that catalyzes H2Bub (grey circle). H2Aub is replaced by histone variant H2AFZ (pink circle). (**F**) MDC1 recruits NuA4 HAT complex (EPC1-TIP60) that, in association with CBP-p300 HAT complex, induce H4 acetylation on K5, 8 and 12 (yellow-green-light blue circles). (**G**) Dimer H2AFZ/H2Bub facilitate association of MOF HAT to the chromatin for H4K16ac formation as last step of histone H4 hyperacetylation Finally, BRDT, in association to SMARCE1, binds K5ac and K8ac of tetra-acetylated H4 to guide histone removal.

**Figure 6 jcm-09-00640-f006:**
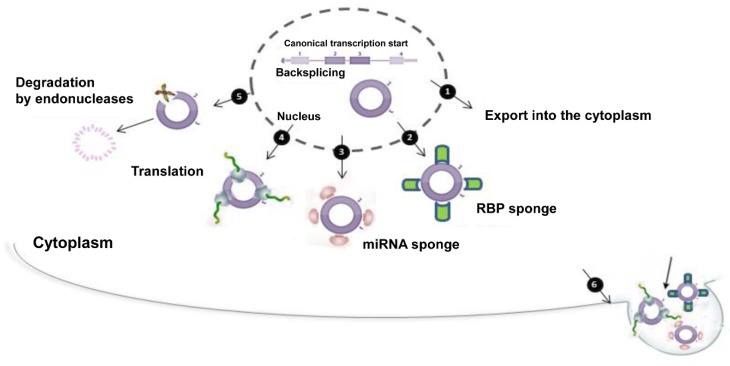
CircRNAs at a glance. Starting from a canonical mRNA, circRNA biogenesis starts through a backsplicing reaction. This event takes place in the nucleus. After that, circRNA may be exported from the nucleus into the cytoplasm (1). In the cytoplasm, circRNA may have multiple actions: it is able to bind to RBPs (2) and/or miRNAs (3) with a typical “sponge activity”; if equipped with IRES sequences, it may be translated (4); circRNA degradation may be possible through endonuclease activity (5); the production of vescicles containing circRNA is a way to remove it from the cytoplasm. After its release into the extracellular space, it probably reaches other cells or tissues perhaps acting as a signal or playing other unknown functions (6).

**Figure 7 jcm-09-00640-f007:**
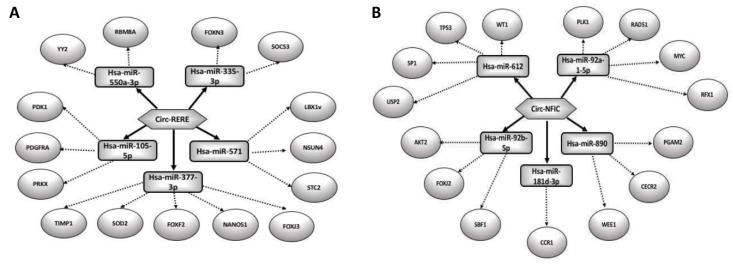
CircRNA-miRNA network analysis. Circ-RERE and circ-NFIC and their predicted miRNAs were selected to generate a network map involved in embryo development (**A**) and germ cell progression (**B**), respectively. The circRNA-miRNA network was constructed using bioinformatic online programs (starBase, circBase, TargetScan, miRBase, Cytoscape).

**Table 1 jcm-09-00640-t001:** Histone PTMs and related histone-modifying enzymes described in testis.

*Enzyme*	*Target Amino Acid*	*Function*	*References*
SUV39H1/SUV39H2	H3K9	Histone methylation	[11]
G9a	H3K9	Histone methylation	[12]
DOT1L	H3K79	Histone methylation	[13]
MLL2	H3K4	Histone methylation	[14]
PRMD9	H3K4	Histone methylation	[15]
SETD2	H3K36	Histone methylation	[16]
ESET	H3K9	Histone methylation	[17]
KDM3A	H3K9	Histone demethylation	[18,19]
TRIP	H3K9	Histone demethylation	[20]
FBXL10	H3K4/K36	Histone demethylation	[21]
LSD1	H3K4	Histone demethylation	[22]
KDM4D	H3K9	Histone demethylation	[23]
TIP60	H4/H2A	Histone acetylase	[24]
MOF	H4K16	Histone acetylase	[25]
SIRT1	H4K16	Histone deacetylase	[26]
RNF8	H2A/H2B	Histone ubiquitine E3	[27]
RNF20	H2B	Histone ubiquitine E3	[28]
HR6B	H2A	Histone ubiquitine E2	[29]
UBR2	H2A	Histone ubiquitine E3	[30]
TSSK6	H2A.X	Histone kinase	[31]
PARP1/2	H1/H2B	Histone ADP ribosyl-transferase	[32]
UBC9	H3/H4/H2A/H2B	Sumo-conjugating enzyme	[33]
PHF7	H2A	Histone ubiquitine E3	[34]
CDYL	H3/H4/H2A/H2B	Histone Crotonyl-CoA Hydratase	[35]

**Table 2 jcm-09-00640-t002:** The main functions of sperm RNA types in spermatogenesis and SPZ.

*RNA Types*	*Functions in Spermatogenesis*	*Functions in SPZ*
*miRNAs*	MiRNAs are involved in the maintenance of pluripotency in germ cells [59] MiRNAs might contribute to early spermatogonial differentiation [60] MiRNAs are preferentially expressed in germ cells during meiosis [61] By regulating the expression of TNP2, they control histone replacement during spermiogenesis [62]	MiRNA reduction in SPZ is associated with subfertility [63] MiRNAs dynamically regulate epididymal environment [64] SPZ-borne miRNAs modulate maternal transcripts prior to activation of the zygotic genome and regulate the first cell division of mouse embryos [65] Different miRNA profiles in sperm samples with high levels of abnormal morphology and low motility [66] Several genes encoding proteins involved in sperm structure, motility and metabolism are miRNA targets [67] Diet-induced paternal obesity influences sperm miRNA content [68]
*piRNAs*	Pre-pachytene piRNAs (5%) are expressed until the onset of meiosis, pachytene piRNAs are expressed in SPC and round SPT [69] In pre-pachytene phase, they guide MILI and MIWI2 proteins to cleave transposons [70] In murine SPC and SPT, piRNAs map to specific genomic regions [71] Mouse mutants defective in piRNA pathway exhibit meiotic arrest at zygotene stage for massive DNA damage [72] Pachytene piRNAs are required for MIWI degradation through the APC proteasome pathway in elongated SPT [73]	Transgenerational effects on gene and transposon expression, as well as heterochromatin changes are associated with the changes of piRNAs and PIWI proteins expression in sperm [74] SPZ-borne piRNAs may influence gene expression of embryos [75] Dietary folate supplementation changes piRNA profiles in SPZ [76] In the silencing of mobile genetic elements piRNAs are considered as potential mediators of epigenetic transgenerational inheritance [77]
*tsRNAs*		In testicular sperm, tsRNAs are scarce; they increase as sperm mature in the epididymis [47] tsRNAs are essential for embryonic development in mice [4] Dietary sugar acutely modulates tsRNAs in sperm; diet sensitive tsRNAs correlate with sperm motility [78]
*rsRNAs*		Mature SPZ are enriched in rsRNAs [50] rsRNAs are considered sperm quality biomarkers [49] Small RNA profiles in ejaculated sperm, epididymal sperm and seminal plasma indicate that sperm RNAs may have origins other than the testes [79] The deletion of a mouse tRNA methyltransferase, *DNMT2*, abolisheds sperm sncRNA-mediated transmission of high-fat diet (HFD)-induced metabolic disorders to offspring [80]

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
