# Peer review of "Histone Post-Translational Modifications and CircRNAs in Mouse and Human Spermatozoa: Potential Epigenetic Marks to Assess Human Sperm Quality"

_jcm, 2020, doi:10.3390/jcm9030640_

Round 1

Reviewer 1 Report

General comments:

The review manuscript by Chioccarelli et al. provides an overview of the current scientific knowledge regarding the epigenetic function of posttranslational histone modifications and of cirRNAs for spermatogenesis and sperm quality, and their contribution to fertilization success and embryo development. The review is very detailed and provides an exhaustive overview of the current literature. It is timely and relevant to the field of reproductive medicine and reproductive science in general.

Comments and suggestions:

Language:

In order to help with the readability of the manuscript, it would be advisable to have the manuscript read and corrected by a native English speaker. There are frequent minor grammatical errors and the incorrect usage of some verbs at times impedes the readability of the manuscript. General example for areas language correction that would be necessary are:

Use of articles when appropriate. Often articles are missing. Use of plural forms of nouns when not appropriate (e.g. “evidences” – there is no plural form of the noun evidence in the English language) Incorrect use of prepositions Incorrect use of the term “thus” with infinitive vs. gerund Have usage of a word like “instead”, “however”, etc. double-checked by a native speaker The term “results to be” is frequently and incorrectly used throughout the manuscript, hindering the comprehensibility of many sentences. It might be intended to substitute the somewhat colloquial expression “turns out to be”.

Here are a few examples taken from the beginning of the manuscript. Correcting all mistakes would exceed the purpose and scope of a manuscript review.

L.40, 54, 62, 72: The term “mentioned” is confusing. Probably “designated” would be a better word.

L 69-71. The meaning of this sentence is unclear. What does “seems to be liked to” mean in this context?

75: replace “mammalians” with “mammals” 85: Check the spelling of crotonylase 354: ALDH2 and STRA8 should not be labeled as “meiotic genes” 102: replace “deeply explained”, e.g. with “sufficiently explained” 114: correct spelling of emphasized

L.373: delete “useful”

Figures:

Fig. 1: Size of labels has to be increased significantly, very hard to read at the current size

Fig. 2: References or citations should be given. Some terms are confusing: Why are some PTMs given as general unspecific terms (H4Ac or H3ac), when at different steps more specific information is given, e.g. H4K16ac? Also, the scientific literature does not fully support an association of H4K16ac with meiotic cells but seems to agree on H4K16ac presence.

Fig. 6: Increase the font size of labels.

Fig. 7: The whole figure needs to be increased significantly since at the current size the labels are illegible.

Content-related comments:

Section 1.1 and 1.2: The insertion of tables that would provide overviews of the various histone PTMs (section 1.1) and RNA types (section 1.2) would be helpful. The section on RNA species seems unbalanced. For a review on spermatogenesis and spermatozoa, it would be useful for the reader to be provided with an overview section over the current knowledge on the roles of the various other RNA types that are known to have roles in spermatogenesis had spermatozoa. The introductory section on RNA types seems misplaced in between the intro on histone PTMs and the exhaustively detailed section on PTMs. A detailed introduction into sperm RNA types (including an overview table with information on what is known regarding their function for spermatogenesis and spermatozoa) in connection with the detailed section on circRNAs might help. The circRNA section seems at times too exhaustive and speculative for a scientific review article, given the currently limited knowledge on the function of circRNAs in this context. Line 372, Lies 406-410, Fig.4: It is not accurate to state that H4K16 is acetylated in preleptotene cells. Also, it is not suitable to cite papers describing results obtained in yeast, but claim the results happen in “mouse and human”. Line 408: “In germ cell, the loss of histone deacetylase MOF causes global loss of H4K16ac” – please insert a reference that demonstrates that this histone modification (i.e. H4K16ac) is occurring in mammalian spermatogenesis (and that it can be visualized in whole testis sections); otherwise, please reword and adjust figure 4 accordingly. The cited references do not corroborate this statement and mechanisms. The current literature seems to focus on the role of this mechanism in DNA-damage mediated repair processes. In order to include this mechanism in meiotic homologous recombination, please insert appropriate literature references that demonstrate the occurrence and role of H4K16ac in meiotic DNA recombination. Line 645-646: The references cited only analyze sperm chromatin packaging, but do not analyze embryonic gene expression. A reference connecting information of altered sperm genome to altered embryonic gene expression should be cited to substantiate the statement. Line 354: ALDH2 should not be classified as a meiotic gene. Line 418: Include explanation MSCI (meiotic Sex Chromosome Inactivation) Line 694: Reference 167 describes a study performed in Xenopus laevis. In this species, sperm chromatin is organized somewhat differently from the mammalian sperm chromatin. It would be appropriate to either add the info that this information refers to Xenopus, or to include a reference to a study performed in mammalian sperm. Lines 797-800: Introduce what the abbreviations stand for (QKI, MBL, FUS) 31 ff: It should be made clear that general terms like histone acetyl transferases (HATs), HDAcs, HMTs and HDMs describe members of larger enzyme families, not one specific enzyme

Author Response

Reviewer 1

General comments:

The review manuscript by Chioccarelli et al. provides an overview of the current scientific knowledge regarding the epigenetic function of posttranslational histone modifications and of cirRNAs for spermatogenesis and sperm quality, and their contribution to fertilization success and embryo development. The review is very detailed and provides an exhaustive overview of the current literature. It is timely and relevant to the field of reproductive medicine and reproductive science in general.

Comments and suggestions:

Language:

In order to help with the readability of the manuscript, it would be advisable to have the manuscript read and corrected by a native English speaker. There are frequent minor grammatical errors and the incorrect usage of some verbs at times impedes the readability of the manuscript. General example for areas language correction that would be necessary are:

Use of articles when appropriate. Often articles are missing. Use of plural forms of nouns when not appropriate (e.g. “evidences” – there is no plural form of the noun evidence in the English language) Incorrect use of prepositions Incorrect use of the term “thus” with infinitive vs. gerund Have usage of a word like “instead”, “however”, etc. double-checked by a native speaker The term “results to be” is frequently and incorrectly used throughout the manuscript, hindering the comprehensibility of many sentences. It might be intended to substitute the somewhat colloquial expression “turns out to be”.

We thanks reviewer for his consideration. The manuscript has been completely revised by a native English speaker.

Here are a few examples taken from the beginning of the manuscript. Correcting all mistakes would exceed the purpose and scope of a manuscript review.

-L 40, 54, 62, 72: The term “mentioned” is confusing. Probably “designated” would be a better word.

We thanks reviewer for his consideration. We changed the text of manuscript, accordingly.

-L 69-71. The meaning of this sentence is unclear. What does “seems to be liked to” mean in this context?

We thanks reviewer for his consideration. Actually there has been a typo; the correct sentence is “seems to be linked to”. We changed the text of manuscript, accordingly.

-L 75: replace “mammalians” with “mammals”

We changed the text of manuscript, accordingly.

-L 85: Check the spelling of crotonylase

We changed the text of manuscript, accordingly.

-L 354: ALDH2 and STRA8 should not be labeled as “meiotic genes”

We thanks reviewer for his consideration. We changed the text of manuscript, accordingly.

-L 102: replace “deeply explained”, e.g. with “sufficiently explained”

We changed the text of manuscript, accordingly.

-L 114: correct spelling of emphasized

We changed the text of manuscript, accordingly.

-L 373: delete “useful”

We changed the text of manuscript, accordingly.

Figures:

Fig. 1: Size of labels has to be increased significantly, very hard to read at the current size

Fig. 2: References or citations should be given. Some terms are confusing: Why are some PTMs given as general unspecific terms (H4Ac or H3ac), when at different steps more specific information is given, e.g. H4K16ac? Also, the scientific literature does not fully support an association of H4K16ac with meiotic cells but seems to agree on H4K16ac presence.

Fig. 6: Increase the font size of labels.

Fig. 7: The whole figure needs to be increased significantly since at the current size the labels are illegible.

We thanks reviewer for his considerations. We changed the figures, accordingly.

Content-related comments:

Section 1.1 and 1.2: The insertion of tables that would provide overviews of the various histone PTMs (section 1.1) and RNA types (section 1.2) would be helpful. The section on RNA species seems unbalanced. For a review on spermatogenesis and spermatozoa, it would be useful for the reader to be provided with an overview section over the current knowledge on the roles of the various other RNA types that are known to have roles in spermatogenesis had spermatozoa. The introductory section on RNA types seems misplaced in between the intro on histone PTMs and the exhaustively detailed section on PTMs. A detailed introduction into sperm RNA types (including an overview table with information on what is known regarding their function for spermatogenesis and spermatozoa) in connection with the detailed section on circRNAs might help. The circRNA section seems at times too exhaustive and speculative for a scientific review article, given the currently limited knowledge on the function of circRNAs in this context.

We thanks reviewer for this comment. As suggested, two tables providing an overview of the various histone PTMs (section 1.1) and RNA roles in both spermatogenesis and spermatozoa (section 1.2) have been added, accordingly. However, a detailed description of sperm RNA types beyond circRNAs in the introduction section has been considered, in our opinion, away from the focus of this review that, rather, has tried to emphasize the novel function of circRNAs in the field of reproduction, despite the limited knowledge in such a context.

In addition, section 1.2 is not misplaced between histone PTMs and the exhaustively detailed section of PTMs, rather it is an overview of recent discoveries concerning several classes of RNAs in spermatozoa that, together with a preliminary view of histone code, anticipates to the detailed sections.

-Line 372, Lies 406-410, Fig.4: It is not accurate to state that H4K16 is acetylated in preleptotene cells. Also, it is not suitable to cite papers describing results obtained in yeast, but claim the results happen in “mouse and human”.

We thanks reviewer for his consideration. We changed the text of manuscript, accordingly.

-Line 408: “In germ cell, the loss of histone deacetylase MOF causes global loss of H4K16ac” – please insert a reference that demonstrates that this histone modification (i.e. H4K16ac) is occurring in mammalian spermatogenesis (and that it can be visualized in whole testis sections); otherwise, please reword and adjust figure 4 accordingly. The cited references do not corroborate this statement and mechanisms. The current literature seems to focus on the role of this mechanism in DNA-damage mediated repair processes. In order to include this mechanism in meiotic homologous recombination, please insert appropriate literature references that demonstrate the occurrence and role of H4K16ac in meiotic DNA recombination.

We thanks reviewer for his consideration. We inserted new references and changed the test of manuscript, accordingly.

- Line 645-646: The references cited only analyze sperm chromatin packaging, but do not analyze embryonic gene expression. A reference connecting information of altered sperm genome to altered embryonic gene expression should be cited to substantiate the statement.

We thanks reviewer for his consideration. We inserted a new reference, accordingly.

- Line 354: ALDH2 should not be classified as a meiotic gene.

We changed the test of manuscript, accordingly.

-Line 418: Include explanation MSCI (meiotic Sex Chromosome Inactivation)

The explanation of the MSCI acronym has already been provided (see line 380-381 of new version of manuscript).

- Line 694: Reference 167 describes a study performed in Xenopus laevis. In this species, sperm chromatin is organized somewhat differently from the mammalian sperm chromatin. It would be appropriate to either add the info that this information refers to Xenopus, or to include a reference to a study performed in mammalian sperm.

We thanks reviewer for his consideration. We have changed the old reference 167 and replaced it with one that refers to studies performed in mammalian sperm, in accord to reviewer request.

- Lines 797-800: Introduce what the abbreviations stand for (QKI, MBL, FUS)

We changed the test of manuscript, accordingly.

-31 ff: It should be made clear that general terms like histone acetyl transferases (HATs), HDAcs, HMTs and HDMs describe members of larger enzyme families, not one specific enzyme 

We thanks reviewer for his consideration. We changed the test of manuscript, accordingly.

Reviewer 2 Report

The manuscript provides a good review of epigenetic mechanisms in spermatozoa with main focus on post translational modifications and circRNA. Further, the authors discuss the role of these epigenetic mechanisms in the assessment sperm quality. The manuscript provides a detail information about the topic but the English requires extensive editing.

Minor concerns

The quality of some figures in the manuscript has to be greatly improved and text fonts made larger for example Figure 7. The writings are very small and extremely difficult to read.

In the Glossary I think it will be great if the authors write Histone before the PTMs e.g Histone acetylation, histone methylation etc because other proteins are acetylated for example which are not histones.

Line 36: Replace Typic with Typical

Line 47: Replace demethylase with demethylases

Line 70: Linked not liked

Line 91 ……outside the nucleus instead poly-ADP ribosylation, What does this mean?

Line 105: Replace is referred as to “is referred to as the”

Line 123 the title Histone PTMs and histone code should be replaced to “Histone code and Histone PTMs).

Line 131-132………being DNA negatively charged should be changed to “DNA being negatively charged”.

Line 236: replace primary with primarily

Line 143: Replace consequentially with consequently

Line 146: association with not of protein

Line 165 constitute a not an heterogenous

Line 172: tRNAs can be classified into not in

Line 191-192 the sentence does not make sense. Revise the English

Line 220: “considered as involved molecules” Revise the English

Line 236: “has had”, Revise

Line 266: hallmark should be hallmarks

Line 286: to creating?

Line 293-294: control of this activity maintenance makes no sense

Line 295: start sites not start site

Line 296: are less present?

Line 373-134: Do the authors mean hotspot or hotpots? This has to be corrected in the entire manuscript especially in the legend of Figure 4.

Line 432: As a result not as result

Line 443: ontoso and colleagues not collogues

Line 594: increase in SPZ not increase of SPZ

Line597: Trichostatin A not trichostatine A

Line 625: any tissue or cell line not any tissues or cell lines

Line 637: correlated with protamine not to

Line 763: What is known about …..not what it is known

Line 1056: man fertility or male fertility?

References: why are some references underlined and others not?

Author Response

Reviewer 2

Comments and Suggestions for Authors

The manuscript provides a good review of epigenetic mechanisms in spermatozoa with main focus on post translational modifications and circRNA. Further, the authors discuss the role of these epigenetic mechanisms in the assessment sperm quality. The manuscript provides a detail information about the topic but the English requires extensive editing.

We thanks reviewer for his considerations. The manuscript has been completely revised by a native English speaker.

Minor concerns

-The quality of some figures in the manuscript has to be greatly improved and text fonts made larger for example Figure 7. The writings are very small and extremely difficult to read.

We thanks reviewer for his considerations. We changed the figures as suggest also by reviewer 1.

-In the Glossary I think it will be great if the authors write Histone before the PTMs e.g Histone acetylation, histone methylation etc because other proteins are acetylated for example which are not histones.

We thanks reviewer for his considerations. We changed the text of glossary, accordingly

Line 36: Replace Typic with Typical

Line 47: Replace demethylase with demethylases

Line 70: Linked not liked

Line 91 ……outside the nucleus instead poly-ADP ribosylation, What does this mean?

Line 105: Replace is referred as to “is referred to as the”

Line 123 the title Histone PTMs and histone code should be replaced to “Histone code and Histone PTMs).

Line 131-132………being DNA negatively charged should be changed to “DNA being negatively charged”.

Line 236: replace primary with primarily

Line 143: Replace consequentially with consequently

Line 146: association with not of protein

Line 165 constitute a not an heterogenous

Line 172: tRNAs can be classified into not in

Line 191-192 the sentence does not make sense. Revise the English

Line 220: “considered as involved molecules” Revise the English

Line 236: “has had”, Revise

Line 266: hallmark should be hallmarks

Line 286: to creating?

Line 293-294: control of this activity maintenance makes no sense

Line 295: start sites not start site

Line 296: are less present?

Line 373-134: Do the authors mean hotspot or hotpots? This has to be corrected in the entire manuscript especially in the legend of Figure 4.

Line 432: As a result not as result

Line 443: Ontoso and colleagues not collogues

Line 594: increase in SPZ not increase of SPZ

Line597: Trichostatin A not trichostatine A

Line 625: any tissue or cell line not any tissues or cell lines

Line 637: correlated with protamine not to

Line 763: What is known about …..not what it is known

Line 1056: man fertility or male fertility?

References: why are some references underlined and others not?

We thanks reviewer for his considerations. All corrections have been made and the text has been changed, accordingly